# Integrating process-related information into an ANN for root-zone soil moisture prediction

Roiya Souissi[1], Mehrez Zribi[1], Chiara Corbari[2], Marco Mancini[2], Sekhar Muddu[3], Sat Kumar Tomer[4], Deepti B Upadhyaya[3,4], Ahmad Al Bitar[1]

[1]CESBIO—Centre d'Etudes Spatiales de la Biosphère, Université de Toulouse, CNES/CNRS/INRAE/IRD/UPS, Toulouse, France

[2]Department of Civil and Environmental Engineering (DICA), Polytechnic University of Milan, 20133 Milano, Italy

[3]Department of Civil Engineering, Indian Institute of Science, Bangalore 560 012, India

[4]Satyukt analytics Pvt Ltd, Sanjay Nagar Main Rd, MET Layout, Bengaluru, Karnataka 560094, India

*Correspondence to*: Roiya Souissi (roiya.souissi@cesbio.cnes.fr)

**Abstract.** Quantification of root-zone soil moisture (RZSM) is crucial for agricultural applications and soil sciences. RZSM impacts processes such as vegetation transpiration and water percolation. Surface soil moisture (SSM) can be assessed through active and passive microwave remote sensing methods, but no current sensor enables direct RZSM retrieval. Spatial maps of RZSM can be retrieved via proxy observations (vegetation stress, water storage change, and surface soil moisture) or via land surface model predictions. In this study, we investigated the combination of surface soil moisture information with process-related inferred features involving artificial neural networks (ANNs). We considered the infiltration process through the soil water index (SWI) computed with a recursive exponential filter and the evaporation process through the evaporation efficiency computed based on a MODIS remote sensing dataset and simplified analytical model, while vegetation growth was not modeled and only inferred through normalized difference vegetation index (NDVI) time series. Several ANN models with different sets of features were developed. Training was conducted considering in situ stations distributed several areas worldwide characterized by different soil and climate patterns of the International Soil Moisture Network (ISMN), and testing was applied to stations of the same data hosting facility. The results indicate that the integration of process-related features into ANN models increased the overall performance over the reference model level in which only SSM features were considered. In arid and semi-arid areas, for instance, performance enhancement was observed when the evaporation efficiency was integrated into the ANN models. To assess the robustness of the approach, the trained models were applied on observation sites in Tunisia, Italy and South-India that are not part of ISMN. The results reveal that joint use of surface soil moisture, evaporation efficiency, NDVI and recursive exponential filter represented the best alternative for more accurate predictions in the case of Tunisia, where the mean correlation of the predicted RZSM based on SSM only sharply increased from 0.443 to 0.801 when process-related features were integrated into the ANN models in addition to SSM. However, process-related features have no to little added value in temperate to tropical conditions.

**Keywords:** root-zone soil moisture, artificial neural networks, evaporation efficiency, exponential filter.

## 1 Introduction

Soil moisture is a major land parameter integrated into several agricultural, hydrological and meteorological applications (Koster et al., 2004; Anguela et al.,2008) This essential climate variable (ECV) consists of two components, namely, surface soil moisture (SSM) (0–5 cm) and root-zone soil moisture (RZSM). RZSM corresponds to the soil moisture in the region in which the main vegetation rooting network is developing. Its definition varies depending on vegetation type and pedoclimatic conditions. The importance of RZSM is mainly highlighted in agricultural applications through vegetation stress and water needs and in carbon and nitrogen cycles, as RZSM influences biogeochemical activities in soil (Martínez-Espinosa et al., 2021). RZSM is nonlinearly related to SSM through different hydrological processes, such as diffusion processes. RZSM may be extracted by evaporation at the surface, through root extraction or by capillary rises (Calvet et Noilhan, 2000). SSM quantification is achieved through three main sources: in situ measurements, model estimates and remote sensing-based products. Microwave remote sensing technologies involving sensors such as the Soil Moisture and Ocean Salinity (SMOS) (Kerr et al., 2010), Soil Moisture Active Passive (SMAP) (Entekhabi et al.,2010) Advanced Microwave Scanning Radiometer (AMSR) (Owe et al., 2008) and Advanced Scatterometer (ASCAT) (Wagner et al., 2013) have been employed to retrieve SSM at coarse resolutions. Current satellite sensors can only provide surface soil moisture information because of the shallow penetration depth of spaceborne data (on the order of a few centimeters) (Wagner et al., 2007). Fine-spatial resolution synthetic aperture radar (SAR) data can also be applied in synergy with optical data to retrieve soil moisture (Zribi et al., 2011; Hajj et al.,2014; Dorigo et al., 2011), but again for surface soil moisture. The International Soil Moisture Network (ISMN) is an exhaustive data hosting facility focused on soil moisture data and associated ancillary information. The ISMN provides in situ soil moisture measurements collected from operational soil moisture networks worldwide (Dorigo et al., 2011). Various models can be adopted to estimate RZSM, such as land surface models (Surfex (Masson et al., 2013), ISBA (Noilhan et al., 1996), CLM (Oleson et al., 2010), JULES (Best et al., 2011), etc.) or dedicated crop models such as Aquacrop (Raes et al., 2009) or SAFYE (Battude et al., 2017). While these models provide the advantage of physical process-based estimates, these estimates depend on the availability and accuracy of ancillary information. Model predictions are often enhanced by the implementation of data assimilation techniques, such as the land data assimilation system (LDAS) (Sabater et al., 2007; Entekhabi et al., 2020).

Data-driven methods such as artificial neural networks (ANNs) have also been commonly applied in hydrology as detailed for instance by the ASCE Task Committee on Application of Artificial Neural Networks in Hydrology (2020) and in (Tanty el al., 2015). One of their advantages is that these models do not require an explicit model structure to accurately represent the involved hydrological processes but instead construct a relationship between the given inputs and the process of interest. Therefore, ANNs are regarded as dynamic input–output mapping models heavily relying on the provided training data relevant to target values (Pan et al., 2017). Moreover, ANNs only require a one-time calibration to provide soil moisture estimations once instrument data are loaded and thus generate relatively low computational costs (Kolassa et al., 2018). These advantages explain the approach to estimate RZSM based on surface information with ANNs in various methodologies (Pan et al., 2017; Grillakis et al., 2021; Souissi et al., 2020). In this paper, we do not address ANN applications as a model twin where the ANN model is trained on the target for mimicking purposes and subsequently generates predictions while requiring a short computation time or fewer input simplifications. Here, we are instead interested in the adoption of ANNs as independent models trained on in situ observations. Within this context, Pan et al. (2017) successfully applied an ANN as a model for shallow 20-cm root zone soil moisture prediction with a global correlation coefficient of 0.7. Grillakis et al. (2021) proposed employing an ANN as a means to calibrate and regionalize the time constant of a recursive exponential filter, which was thereafter applied at the regional scale. A combined implementation of Bayesian probabilistic approach and an ANN to infer

RZSM at different depths from optical UAV acquisitions via local training was also applied (Hassan-Esfahani et al., 2017). Multitemporal averaged features to predict RZSM based on only SSM and investigated the transferability of a trained ANN across different climatic conditions globally were proposed in (Souissi et al., 2020). Temporal information can be considered in ANNs through recurrent neural networks (RNNs), long short-term memory (LSTM) architectures (Liu et al., 2021), 1D convolutional neural networks (CNNs), or multitemporal averaging. In (Souissi et al., 2020), median, maximum, and minimum correlation values of 0.77, 0.96, and 0.65 were respectively reported across training, validation and test datasets. The use of climatic variables such as precipitation and surface temperature and intrinsic surface properties such as soil texture and land cover has also been considered in ANNs (Liu et al., 2021). The choice of variables depends not only on the data availability but also on the objectives. Finally, ANN-based approaches pertain to the more general term of machine learning approaches, and within this framework, the random forest approach has been applied to root zone soil moisture prediction (Carranza et al., 2021). The aforementioned studies have investigated the application of multiple information sources to predict root zone soil moisture. The input features are commonly curated for quality, and correlation analysis is conducted to determine the useful inputs, while physical processes are not considered. In this paper, we introduce process-related features based on simplified analytical models representing the major processes contributing to root zone soil moisture dynamics. In this work, RZSM refers to a point observation of water content in a depth ranging between 30 and 55cm. We investigate the impact of the application of different process-related variables on the precision of RZSM predictions as well as the robustness of our approach. (1) We start from a previously developed ANN model (Souissi et al., 2020), and we extend the feature list to include NDVI time series, surface soil temperature and process-related variables, namely, the soil water index given by a recursive exponential filter and remote sensing-based evaporation efficiency. (2) The robustness of the approach is assessed through additional tests involving stations not included in the ISMN database in Tunisia, Italy, and South-India. (3) Climatic analysis is conducted to infer the most indicative process-related features for each climate pattern.

## 2   Materials and Methods

The proposed methodology entails the construction of several ANN models with both direct (SSM, surface temperature, and NDVI) and intermediate sets of features (soil water index and evaporation efficiency) computed based on simplified analytical models. An overview of the processing configuration is shown in Figure 1. Standard scaling is applied to each dataset separately so that the different inputs fall into the same range of values, then the ANN outputs are descaled to make the comparison with actual values of RZSM possible.

103

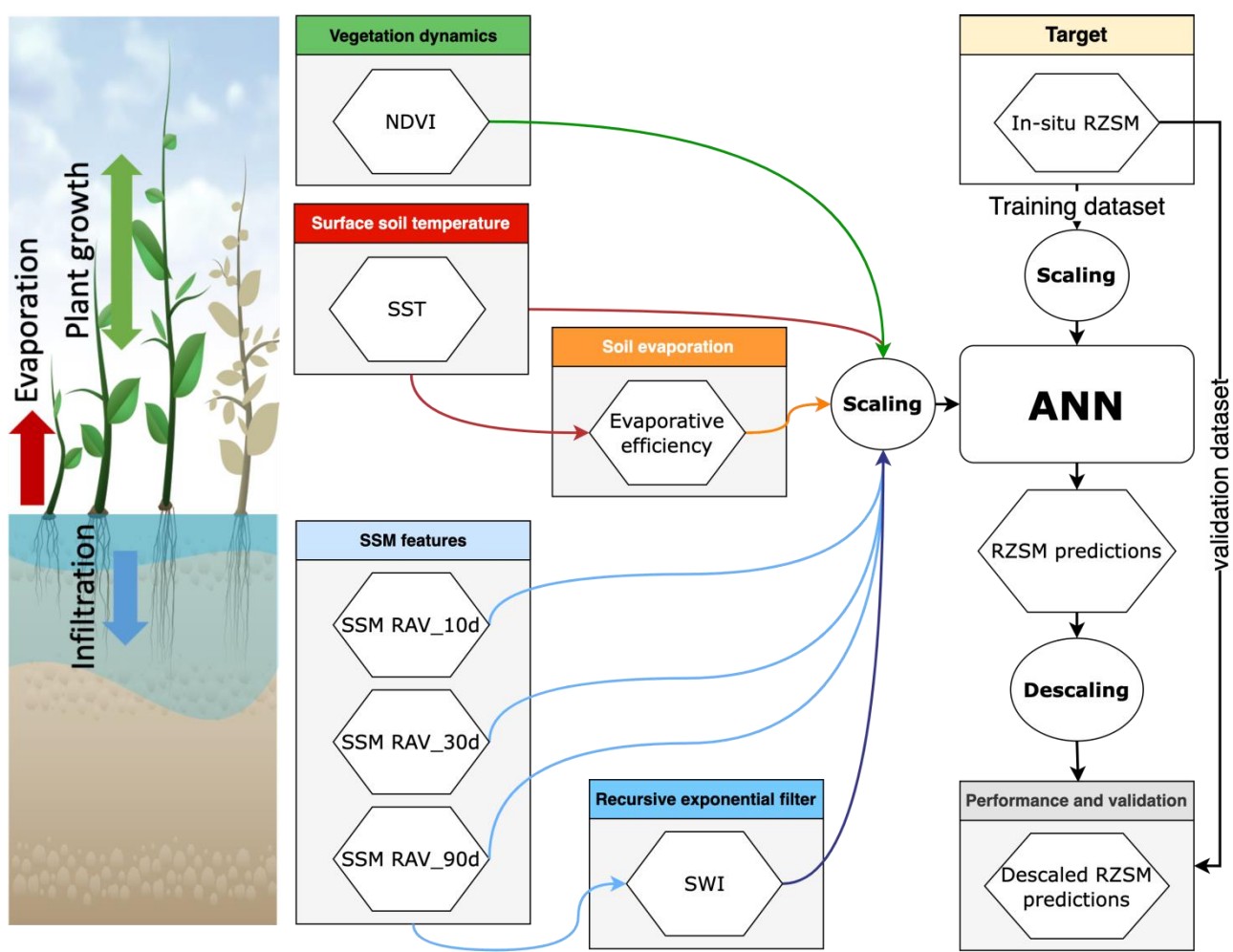

**Figure 1.** Overview of the processing configuration showing the components of the model: the tested models are variations of this ANN with a different combination of inputs (see Table 1). The scaling and descaling are applied to each dataset separately.

This approach results in a combination of ANN models (Table 1). Each model has one or more process-related features in addition to three SSM features which correspond to backward rolling averages of in-situ SSM computed over 10,30 and 90 days. All the ANN model hyperparameters remain the same except the number of input features.

**Table 1.** ANN model configurations with the respective input variables ; *: rolling averages of SSM over 10 days; **: rolling averages of SSM over 30 days; ***: rolling averages of SSM over 90 days; ****: number of parameters of the ANN model.

| Model \ Features | SSM_10d_RAV* | SSM_30d_RAV** | SSM_90d_RAV*** | SST | NDVI | SWI | EVAP | Nb**** |
|---|---|---|---|---|---|---|---|---|
| ANN_SSM | X | X | X | | | | | 101 |
| ANN_SSM_TEMP | X | X | X | X | | | | 121 |
| ANN_SSM_NDVI | X | X | X | | X | | | 121 |
| ANN_SSM_EXP-FILT-T5 | X | X | X | | | X | | 121 |
| ANN_SSM_EVAP- | X | X | X | | | | X | 121 |

| | | | | | | | |
|---|---|---|---|---|---|---|---|
| EFF-B60 | | | | | | | |
| ANN_SSM_NDVI_E VAP-EFF-B60_EXP-FILT-T5 | X | X | X | X | X | X | 161 |

The model with the simplest starting point is ANN_SSM based on (Souissi et al., 2020). The most complex model includes the full set of inputs. Intercomparison of the model performance provides information on the added value of each input. All input features are scaled, and training is performed on each of these features based on scaled in situ RZSM data retrieved from the ISMN. The RZSM model predictions are validated against an independent set of observations.

**2.1 Datasets**

**2.1.1 ISMN soil moisture data**

The first training and test operations were conducted on eight ISMN networks previously considered in (Souissi et al., 2020). Figure 2 shows the distribution of the considered soil moisture networks with different soil textures and climatic parameters (cf. appendix B). For each station, the RZSM observation point is located between 30 and 55cm (Table 2). For each soil moisture hourly acquisition, ISMN provides quality flags. Quality flags can be marked as 'C' (exceeding plausible geophysical range),' D' (questionable/dubious), 'M' (missing), or 'G' (good) (Dorigo et al.,2011). Category 'D' has subset flags namely 'D01' for which in situ soil temperature < 0°C, 'D02' that flags points at which in situ air temperature < 0°C as well as 'D03' that also flags areas where GLDAS soil temperature < 0°C. In our study, only soil moisture data which quality flag is marked 'G' were retained.

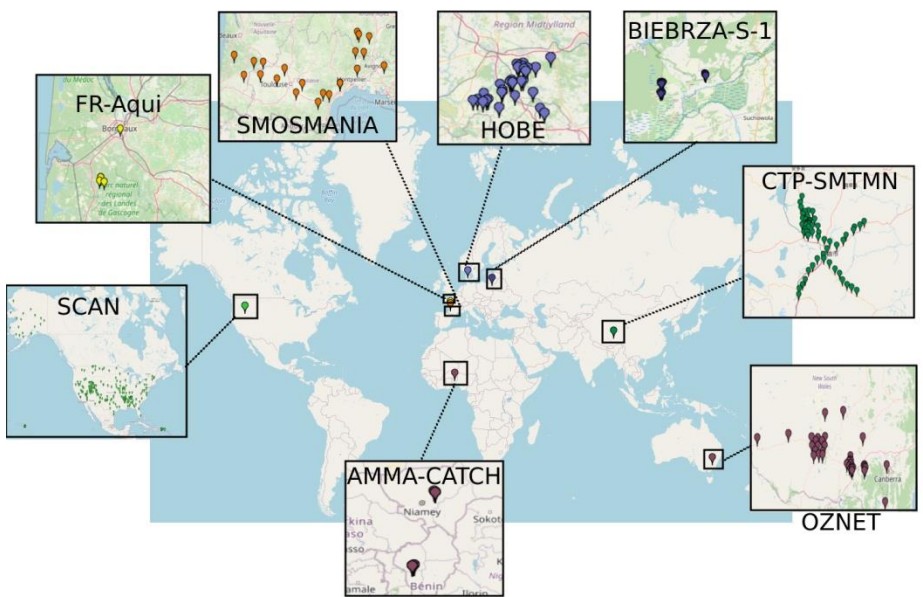

**Figure 2.** International Soil Moisture Network (ISMN) network distribution (adapted from the ISMN web data portal (https://www.geo.tuwien.ac.at/insitu/data_viewer/); scale: 1 cm=1000 km).

**Table 2.** Overview of the considered ISMN and external networks.

| Network | Country | Number of Selected Stations | Selected RZSM Depth (cm) | SM Sensors |
|---|---|---|---|---|
| AMMA-CATCH | Benin, Niger | 5 (3 in Benin and 2 in Niger) | 40 | CS616 |
| BIEBRZA-S-1 | Poland | 3 | 50 | GS-3 |
| CTP-SMTMN | China | 54 | 40 | EC-TM/5TM |
| HOBE | Denmark | 29 | 55 | Decagon-5TE |
| FR-Aqui | France | 5 | 30, 34, 50 | ThetaProbe ML2X |
| OZNET | Australia | 19 | 30 | Hydra Probe-CS616 |
| SCAN | USA | 209 | 50 | Hydraprobe-Sdi-12/Ana |
| SMOSMANIA | France | 22 | 30 | ThetaProbe ML2X |

### 2.1.2 External soil moisture data

The external networks only considered to assess the transferability and robustness of the approach were employed for validation. The trained models are run for predictions only over these sites. They have been selected to cover semi-arid, moderate and tropical semi-arid climates.

- Tunisian site: The Merguellil site is located in central Tunisia (9°54 E; 35°35 N). This site is characterized by a semiarid climate with highly variable rainfall patterns (average equal to 300mm/year), very dry summer seasons, and wet winters. The Merguellil site represents an agricultural region where croplands, namely, olive groves and cereal fields, prevail (Zribi et al., 2021). At this study site, a network of continuous thetaprobe stations installed at bare soil locations provided moisture measurements at depths of 5 and 40 cm. All measurements were calibrated against gravimetric estimations. Data were obtained from the Système d'Information Environmental (SIE) web application catalog.

- Italian site: The Landriano site is located in northern Italy (Pavia province, Lombardia region). This station is located in a maize field, which was monitored in 2006 and from 2010 to 2011 (Masseroni et al., 2014). The average rainfall in Pavia province is of 650–700 mm, the climate is classified as 'Cfa' (cf. appendix A) and the field is irrigated by the border method with an average irrigation amount of approximately 100 to 200 mm per application with one to two applications per season due to the presence of a shallow groundwater table. Soil moisture measurements were performed with time domain reflectometer (TDR) soil moisture sensors. Five TDR soil moisture sensors were installed along a profile at depths of 5, 20, 35, 50, and 70 cm.

Indian site: The Berambadi watershed is located in Gundalpet taluk, Chamarajanagara district, in the southern part of Karnataka state in India and covers an area of approximately 84 km². The average rainfall is equal to 800 mm/year and the climate is classified as Aw (cf. appendix A). Hydrological variables have been intensively monitored since 2009 in the Berambadi watershed by the Environmental Research Observatory ORE BVET and AMBHAS Observatory. The soil moisture levels at the surface (5 cm) and root zone (50 cm) are monitored

with a HydraProbe sensor at different agricultural sites across the watershed, and in the current study, 4 stations
were chosen.
**2.1.3 Surface soil temperature**
In addition to in situ soil moisture, the ISMN optionally includes meteorological and soil variables that are available
over specific time periods. Values of the situ surface soil temperature among these variables can be employed as a
useful indicator of the soil moisture data quality. The soil temperature was provided in Celsius, and the plausible values
range from -60 to 60 °C. Regarding soil moisture data, surface soil temperature data were also provided with quality
flags (Dorigo et al., 2011). However, the drawback is that this variable is not available in all networks, which is the
case with the AMMA-CATCH network.
**2.1.4 Normalized difference vegetation index**
We considered the remote sensing-based normalized difference vegetation index (NDVI) to infer vegetation dynamics.
We extracted this index from the Moderate Resolution Imaging Spectroradiometer (MODIS) Vegetation Indices
product (MOD13Q1 version 6). MODIS Vegetation Indices (MOD13Q1) version 6 data are generated at 16-day
intervals and a 250-m spatial resolution as a Level 3 product. This product provides two primary vegetation layers. The
first vegetation layer is the NDVI, which is referred to as the continuity index of the existing National Oceanic and
Atmospheric Administration-Advanced Very High Resolution Radiometer (NOAA-AVHRR)-derived NDVI. The
algorithm chooses the best available pixel value from all the acquisitions over the 16-day period. The criteria
considered are low cloud coverage, low view angle, and highest NDVI value (Huete et al., 1999). To obtain daily
NDVI values, we conducted linear interpolation of the 16-day product.
**2.1.5 Potential evapotranspiration**
Similarly, we assessed the impact of considering a remote sensing-based evaporation efficiency, which is initially
defined as the ratio of actual to potential soil evaporation, on RZSM prediction. The computation details of this variable
will be detailed later (cf. Section 2.2.2). We employed the remote sensing-based potential evapotranspiration (PET) to
compute the evaporation efficiency. We extracted the PET from the MOD16A2 Evapotranspiration/Latent Heat Flux
version 6 product, which is an 8-day composite dataset produced at a 500-m pixel resolution. The algorithm used for
MOD16 data product collection is based on the logic of the Penman–Monteith equation, which employs inputs of daily
meteorological reanalysis data along with MODIS remote sensing data products such as vegetation property dynamics,
albedo, and land cover. The MOD16A2 product provides layers for the composite evapotranspiration (ET), latent heat
flux (LE), potential ET (PET) and potential LE (PLE). The pixel values for the PET layer include the sum of all eight
days within the composite period (Running et al., 2017). To obtain daily PET values, we performed a linear
interpolation over the 8-day product and then we divided by eight the interpolated value.
**2.2 Methods**
**2.2.1 Recursive exponential filter**
Two ANN models presented in Table 1 contained extra knowledge on infiltration process information based on the
outputs of the recursive exponential filter (Stroud, 1999) as a feature. The recursive exponential filter was first

introduced by Wagner et al. (1999) to estimate the soil water index (SWI) from surface soil moisture. SWI is computed as follows:

$$SWI_{t_n} = SWI_{t_{n-1}} + K_n(ms(t_n) - SWI_{t_{n-1}}) \quad (1)$$

where:

- SWI$_{t_n}$ is the soil water index at time $t_n$,
- ms($t_n$) is the scaledsurface soil moisture at time $t_n$ (scaled between maximum and minimum values),
- $K_n$ is the gain at time $t_n$, which occurs in [0,1] and is given by:

$$K_n = \frac{K_{n-1}}{K_{n-1} + e^{-\frac{(t_n - t_{n-1})}{T}}} \quad (2) \text{ and}$$

- T is a time constant and is the only required tuning parameter to compute the recursive exponential filter.
- For the initialisation of the filter, gain $K_1 = 1$ and $SWI_{(t1)}^* = ms(t_1)$

Regarding T values, we considered an empirical list ([1,3,5,7,10,13,15,20,40,60]), which was partly inspired by (Paulik et al., 2014) (T ∈ [1,5,10,15,20,40,60,100]). Given the list of T values, recursive exponential filter outputs were computed for all of the stations (346 stations) given each T value. Based on the correlation values between the in situ RZSM values and the recursive exponential filter-based RZSM pre-estimates, we established the optimal time variable T, hereafter referred to as T$_{best}$, for each station.

## 2.2.2 Evaporative efficiency

An ANN model with evaporation efficiency input was also developed. This variable, which is defined as the ratio of the actual to potential soil evaporation, was first introduced in (Noilhan, J. and Planton, 1989; Jacquemin et al., 1990; Lee et al., 1992) and thereafter readapted in (Merlin et al., 2010) to include the soil thickness. In our work, we use a modified evaporation efficiency formulation, based on the third model developed in (Merlin et al., 2010), which can be expressed as follows (cf. appendix C):

$$\beta = [\frac{1}{2} - \frac{1}{2}\cos(\pi\theta/\theta_{max})]^{P*} \quad (3)$$

where: - $\beta$ is evaporation efficiency

- $\theta$ is the water content in the soil layer of thickness L.

- $\theta_{max}$ is the maximum soil moisture at each station.

- P$^*$ is a parameter computed as follows:

$$P^* = \frac{PET}{2B} \quad (4)$$

P*, a proxy of parameter P (cf. appendix C), represents an equilibrium state controlled by retention forces in the soil, which increase with the thickness L of considered soil and by evaporative demands at the soil surface.

-PET is the potential evapotranspiration (PET) extracted from the MODIS 500-m 8-day product (MOD16A2).

The soil evaporation efficiency computed by model 3, developed in (Merlin et al., 2010), decreases when PET increases. Retention force and evaporative demand make the term P increase (replaced by P*), as if an increase of potential evaporation $LE_p$ (here replaced by PET) at the soil surface would make the retention force in the soil greater. Merlin et al. (2010) tested this approach at two sites in southwestern France using in situ measurements of actual evaporation, potential evaporation, and soil moisture at five different depths collected in summer. Model 3 was able to represent the soil evaporation process with a similar accuracy as the classical resistance-based approach for various soil thicknesses up to 100 cm. Merlin et al. (2010) affirm the parameterization of P as function of $LE_p$ (here PET) indicates that $\beta$ cannot be considered as a function of soil moisture alone since it also depends on potential evaporation. Moreover, the effect of potential evaporation on $\beta$ appears to be equivalent to that of soil thickness on $\beta$. This equivalence is physically interpreted as an increase of retention forces in the soil in reaction to an increase in potential evaporation.

**2.2.3 Artificial neural network implementation**

The multilayer perceptron (MLP), which is a multilayer feed-forward ANN, is one of the most widely applied ANNs, mainly in the field of water resources (Abrahart and See, 2007) The multilayer perceptron contains one or more hidden layers between its input and output layers. Neurons are organized in layers such that the neurons of the same layer are not interconnected and that any connections are directed from lower to upper layers (Ramchoun et al., 2016). Each neuron returns an output based on the weighted sum of all inputs and according to a nonlinear function referred to as the transfer or activation function (Oyebode and Stretch, 2019). The input layer, consisting of SSM values and/or other processrelated variables, is connected to the hidden layer(s), which comprises hidden neurons. The final ANN-derived estimates of the ANN are given by an activation function associated with the final layer denoted as the output layer, based on the sum of the weighted outputs of the hidden neurons.

**3** We started with the ANN model developed in (Souissi et al., 2020), whose architecture consists of one hidden layer of 20 hidden neurons, a tangent sigmoid function as the activation function of the hidden layer, a quadratic cost function as the loss function and the stochastic gradient descent (SGD) technique as the optimization algorithm. This model was developed to estimate RZSM based on only in situ SSM information. SSM was not applied as a feature of hourly values but was employed in the form of three features, namely, SSM rolling averages over 10, 30 and 90 days. Additional ANN models were developed to study, through each model, the impact of the application of the NDVI, SWI, evaporation efficiency and the surface soil temperature as features. A model combining surface soil moisture, NDVI, evaporation efficiency and recursive exponential filter was further considered. These ANN models were trained and validated on the 122 ISMN stations considered of good quality after a data filtering step as detailed in (Souissi et al., 2020).Training of the above ANN models was conducted considering 70% of these 122 stations. Thirty percent was reserved for validation, and testing was conducted at the rest of stations. So in summary, 122 stations were considered for the training/validation of the ANN models and 224 stations, if all input data are available, were used for testing. In a second step, tests were conducted on data external to the ISMN database namely on sites of Tunisia, Italy and India. The trained models over ISMN are used only in prediction mode over these sites. The data for SSM in addition to the other features are used as inputs and RZSM is predicted in outputs.**Results**

**3.1 Exponential filter characteristic time length**

A large proportion of the stations attained an optimal time constant ($T_{best}$) value equal to 60 days which suggests an
abnormally long infiltration time. These stations belong to the SCAN network and exhibit an RZSM acquisition depth
of 50 cm, in contrast other networks such as SMOSMANIA, for instance, where RZSM is retrieved at 30 cm. The high
values correspond to correlation with seasonal dynamics rather than infiltration processes. This depth could explain the
anomalously long infiltration time. This is consistent with (Paulik et al., 2014) in which the average T value with the
highest correlation ($T_{best}$) increased with increasing depth of the in situ observations.
For comparison purposes, Paulik et al. (2014) found that 23.98% of the stations achieved $T_{best}$=5 days, while 21.58% of
the stations achieved $T_{best} \geq 60$ days (60 or 100 days).
Albergel et al. (2008) considered an average $T_{best}$ value of 6 days for the SMOSMANIA network. This value
represented the average $T_{best}$ value for all stations belonging to the SMOSMANIA network. In our case, the average
$T_{best}$ value for all stations of the SMOSMANIA network reached 9 days. In this study, an average $T_{best}$ value could be
established for each station or each network. However, this is not relevant to our work because we aim to evaluate maps
of remote sensing data in next steps, and thus, we did not compute $T_{best}$ at each location. We fixed the value of T to 5
days as a median infiltration time.
**3.2 Intercomparison of the ANN models**
The distribution histograms for training, validation and test stations (Fig. 3) show that the integration of the considered
process-related features improved the prediction accuracy in certain cases compared to the reference. Time series of
good and less good quality of fit were provided in appendix E for training, validation and test stations using reference
model ANN_SSM and the most complex ANN model.

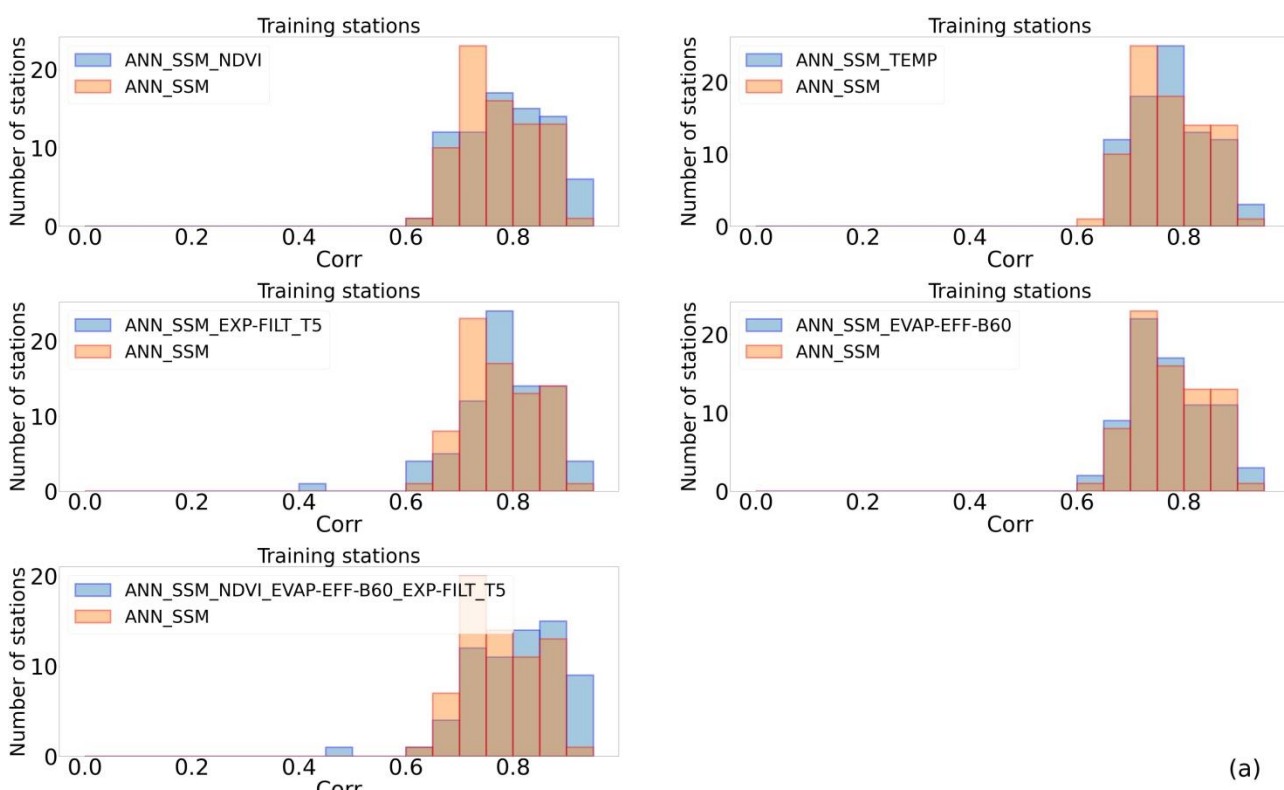


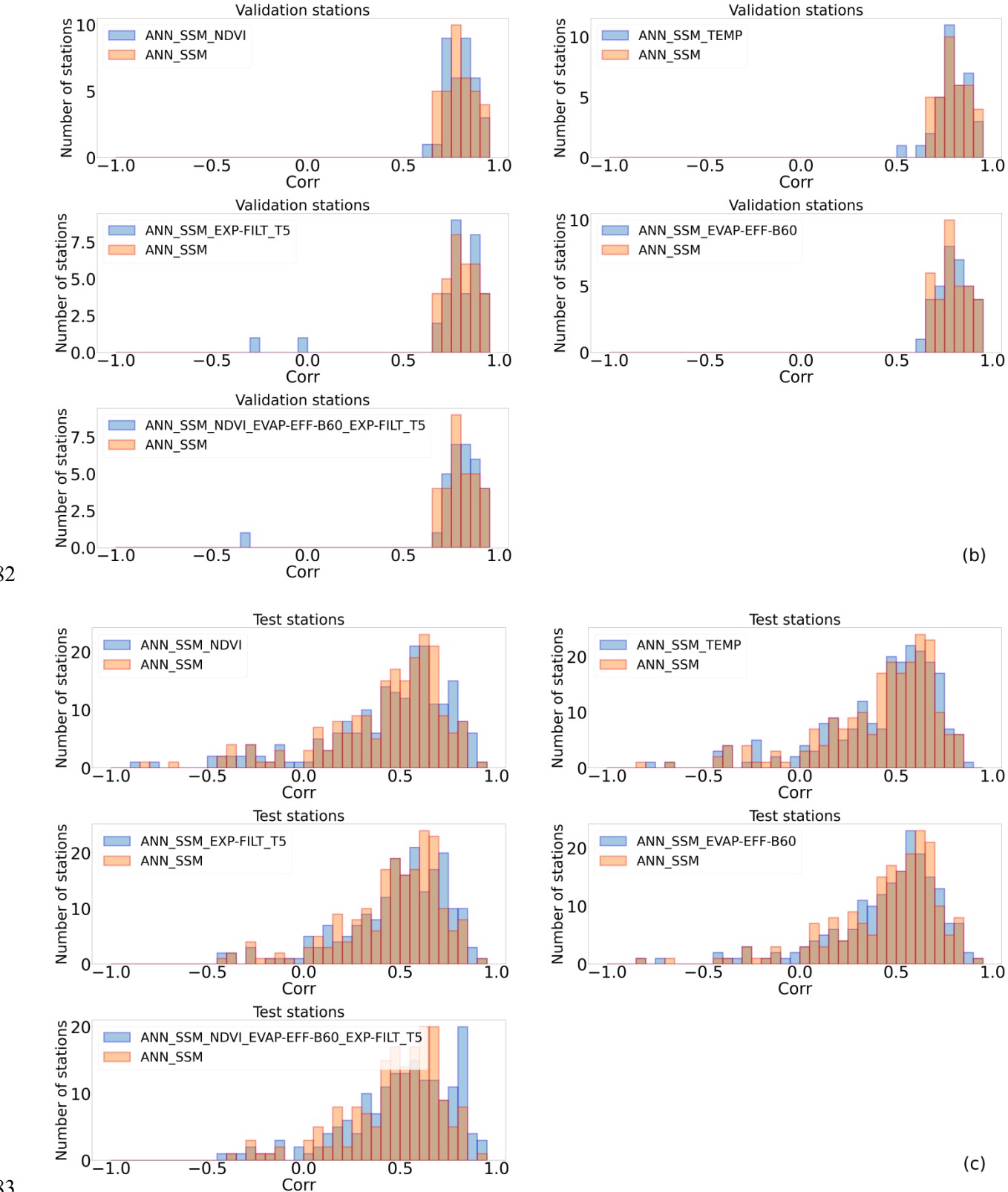



**Figure 3.** Correlation histograms of all tested ANN models compared to ANN_SSM (a) on training stations (b) on validation stations (c) on test stations (cf. appendix D for RMSE histograms)

In terms of the NDVI, 65.82%, 45.71% and 55.22% stations attained better correlation values with ANN_SSM_NDVI than those obtained with ANN_SSM for the training, validation and test stations, respectively. RMSE decreased for

44.3%, 40.0% and 40.3% of the stations with ANN_SSM_NDVI compared to model ANN_SSM for training,
validation and test stations, respectively (Table 3).
In regard to the ANN_SSM_TEMP model that integrates the soil surface temperature, 49.4%, 55.56% and 59.35% of
the training, validation and test stations exhibited higher correlation values than those obtained with the ANN_SSM
model, respectively. RMSE decreased with ANN_SSM_TEMP compared to model ANN_SSM for 25.3%, 38.89% and
42.99% of the training, validation and test stations, respectively.
In addition, model ANN_SSM_EXP-FILT-T5 that integrates the simplified infiltration based features yielded slightly
better correlations, and 64.56%, 60.61% and 63.68% 62.62% of the  training, validation and test stations attained better
correlations than those obtained with model ANN_SSM, respectively. Besides, RMSE decreased for 36.71 %, 42.42 %
and 50.25% of the training, validation and test stations with ANN_SSM_ EXP-FILT-T5 compared to model
ANN_SSM, respectively.
Regarding the evaporation efficiency, we considered different values of fitting parameter B (Eq. 4) such that B
remained within the [50,60] interval. This parameter can be fitted using different variables, such as the wind speed or
relative humidity. Comparisons based on the correlation values provided by the different models for each B value
indicated that the performance was insensitive to the B value. Thus, we fixed the B value to 60 W m$^{-2}$. Comparison of
models ANN_SSM and ANN_SSM_EVAP-EFF-B60 revealed that 54.55%, 52.94% and 52.33% of the training,
validation and test stations attained higher correlation values with the latter model, respectively. RMSE was reduced for
28.57%, 41.18% and 48.19% of the training, validation and test stations with ANN_SSM_ EVAP-EFF-B60 compared
to model ANN_SSM, respectively.
Finally, we investigated the impact of the joint application of the NDVI, recursive exponential filter (T= 5 days)
and  evaporation efficiency (B=60 W m$^{-2}$) in the ANN_SSM_NDVI_EVAP-EFF-B60_EXP-FILT-T5 model. The
surface soil temperature was not included, as its effect is included in the evaporation process. At 84.06%, 61.29% and
62.07% of the training, validation and test stations, the correlation value obtained with this model was higher than that
obtained with the ANN_SSM model, respectively. In addition, RMSE was minimized for 62.32%, 54.84% and 54.02%
of the training, validation and test stations with ANN_SSM_ NDVI_EVAP-EFF-B60_EXP-FILT-T5 compared to
model ANN_SSM, respectively.
Considering model ANN_SSM_NDVI_EVAP-EFF-B60_EXP-FILT-T5, only one training station had a decrease in
correlation by more than 0.1 namely station 'Lind#1' (network 'SCAN') compared to reference model ANN_SSM. All
inputs were not available at the same dates which implied a significant reduction in data points (cf. appendix F). The
decrease in correlation and increase in RMSE didn't exceed 0.1 and 0.01 m$^3$/m$^3$, respectively, for the rest of stations of
lower performance metrics with the most complex ANN.
Similarly for validation stations, only one station had a decrease in correlation above 0.1, namely station 'PineNut'
(network 'SCAN'), with  model  ANN_SSM_NDVI_EVAP-EFF-B60_EXP-FILT-T5. This  decrease  can  be  also
explained because of data shortage (cf. appendix F). The decrease in correlation and increase in RMSE didn't exceed
0.1 and 0.01 m$^3$/m$^3$, respectively, for the rest of stations of lower performance metrics with the most complex ANN.
Regarding test stations , correlation decrease by more than 0.1 and RMSE increase by more than 0.01 m$^3$/m$^3$with model
ANN_SSM_NDVI_EVAP-EFF-B60_EXP-FILT-T5 compared to model ANN_SSM was detected for only 2 stations.

Both stations, namely station 'S-Coleambally' and 'Widgiewa' which belong to network 'OZNET', significantly lose in data volume when process-related variables are integrated in ANN and more precisely because of NDVI data availability (cf. appendix F). For the rest of test stations, correlation decreased and RMSE increased simultaneously by less than 0.1 and 0.01 $m^3/m^3$, respectively, whith model ANN_SSM_NDVI_EVAP-EFF-B60_EXP-FILT-T5.

Table 3. Proportion of the stations which performance enhances using the ANN models enriched with process-related features compared to model ANN_SSM (*: % of stations at which the correlation improves over the model ANN_SSM level; **: % of stations at which RMSE improves over the model ANN_SSM level)

| Model | Training stations | | Validation stations | | Test stations | |
|---|---|---|---|---|---|---|
| | % of stations (corr ↑)* | % of stations (RMSE ↓)** | % of stations (corr ↑)* | % of stations (RMSE ↓)** | % of stations (corr ↑)* | % of stations (RMSE ↓)** |
| ANN_SSM_NDVI | 65.82 | 44.3 | 45.71 | 40.0 | 55.22 | 40.3 |
| ANN_SSM_TEMP | 49.4 | 25.3 | 55.56 | 38.89 | 59.35 | 42.99 |
| ANN_SSM_EXP-FILT-T5 | 64.56 | 36.71 | 60.61 | 42.42 | 63.68 | 50.25 |
| ANN_SSM_EVAP-EFF-B60 | 54.55 | 28.57 | 52.94 | 41.18 | 52.33 | 48.19 |
| ANN_SSM_NDVI_EVAP-EFF-B60_EXP-FILT-T5 | 84.06 | 62.32 | 61.29 | 54.84 | 62.07 | 54.02 |

Table 4. Proportion of the stations which correlation decreases using the ANN models enriched with process-related features compared to model ANN_SSM ($\Delta_{corr}$=corr$_{ANN\_SSM}$ − corr$_{ANN\_SSM\_X}$ , X denotes a or a combination of process-related variables)

| Model | Training stations | | Validation stations | | Test stations | |
|---|---|---|---|---|---|---|
| | % of stations corr ↓ and 0.05<$\Delta_{corr}$<0.1* | % of stations corr ↓ and $\Delta_{corr}$>0.1* | % of stations corr ↓ and 0.05<$\Delta_{corr}$<0.1* | % of stations corr ↓ and $\Delta_{corr}$>0.1* | % of stations corr ↓ and 0.05<$\Delta_{corr}$<0.1* | % of stations corr ↓ and $\Delta_{corr}$>0.1* |
| ANN_SSM_NDVI | 3.8 | 0 | 2.86 | 0 | 9.95 | 5.97 |
| ANN_SSM_TEMP | 0 | 1.2 | 0 | 2.78 | 4.67 | 3.27 |
| ANN_SSM_EXP-FILT-T5 | 6.33 | 1.27 | 3.03 | 9.09 | 6.97 | 3.48 |
| ANN_SSM_EVAP-EFF-B60 | 10.39 | 1.3 | 0 | 2.94 | 6.74 | 5.7 |
| ANN_SSM_NDVI_EVAP-EFF-B60_EXP-FILT-T5 | 4.35 | 1.45 | 6.45 | 3.23 | 9.2 | 6.9 |

Always in terms of the general performance of model ANN_SSM_NDVI_EVAP-EFF-B60_EXP-FILT-T5, about 75% of the stations have an RMSE less than 0.05 $m^3/m^3$ and around half of the stations have an RMSE less than 0.04 $m^3/m^3$. This accuracy is consistent, for instance, with the target value in SMAP (Entekhabi et al., 2010) and SMOS (Kerr et al., 2010) missions which is equal to 0.04 $m^3/m^3$ and also to the average sensor accuracy adopted by Dorigo et al. (2013) which is equal to 0.05 $m^3/m^3$. Overall, the most complex model ANN_SSM_NDVI_EVAP-EFF-B60_EXP-FILT-T5 can successfully characterize the soil moisture dynamics in the root zone since half of the stations have a correlation value greater than 0.7. Pan et al. (2017) developed different ANN models to estimate RZSM at depth of 20cm and 50cm over the continental United States using surface information. They found that half of the stations have RMSE less than 0.06 $m^3/m^3$ and more than 70% of stations have correlation above 0.7 when predicting RZSM at 20cm. However, the developed ANN was less effective in RZSM prediction at 50cm which is also in accordance with (Kornelsen and Coulibaly, 2014). In our study, the densest soil moisture network is 'SCAN', located in the USA. Soil moisture was predicted at a depth of 50cm over this network. Around half of the stations have a correlation value of above 0.6 and RMSE less than 0.04 $m^3/m^3$ after the integration of process-related inputs. Pan et al., (2017) suggests that the use of only time-dependent variables may not be sufficient for the ANN models to accurately predict RZSM and suggests adding soil texture data.

### 3.3 Robustness of the approach

To further assess the robustness of our approach, which involves RZSM prediction using the different ANN models with different features, we predicted RZSM at 40 cm at sites not previously considered in previous parts of the study. The selected stations are located in: the Kairouan Plain, a semiarid region in central Tunisia, Landriano site located in the North of Italy, and the Berambadi watershed located in Gundalpet taluk, South-India. In the case of the Kairouan Tunisia, model ANN_SSM yielded moderate- to low-precision predictions, as highlighted by the performance metrics listed in Table 5. The time series (cf. appendix G) show that the RZSM predictions followed the SSM seasonality, which was reflected by the false peaks generated in the RZSM predictions whenever a sharp increase or decrease occurred in the SSM values. This observation was also found in (Souissi et al., 2020). Actually, the Kairouan Plain is characterized by a semiarid environment where rainfall events infrequently occur and the level of evaporation is high. The reference model ANN_SSM shows its limitations to accurately predict RZSM in areas with no alternate wet and dry cycles.

However, the consideration of additional features, namely, the NDVI, evaporation efficiency and SWI in the ANN models resulted in a good agreement between the in situ and predicted RZSM values (Fig. 4). The correlation values were improved by 60.04%, 169.5%, 112.02%, 80.23% and 53.7% at stations Barrouta-160, Hmidate_163, Barrage_162, Bouhajla_164 and P12, respectively, with the ANN_SSM_NDVI_EVAP-EFF-B60_EXP-FILT-T5 model over ANN_SSM model values. Similarly, RMSE values were reduced (Table 5). As shown in figure 4, the most complex ANN model is able to capture the variations of RZSM. This finding highlights the added value of our hybrid approach based on an association of a machine learning method with process-related variables. Instead of injecting uncertain information in physical models, such as soil properties, we used a nonparametric method related to physical processes without using forcing data that may be subject to errors and potentially lead to inaccurate tracking of the long-term evolution of soil moisture.

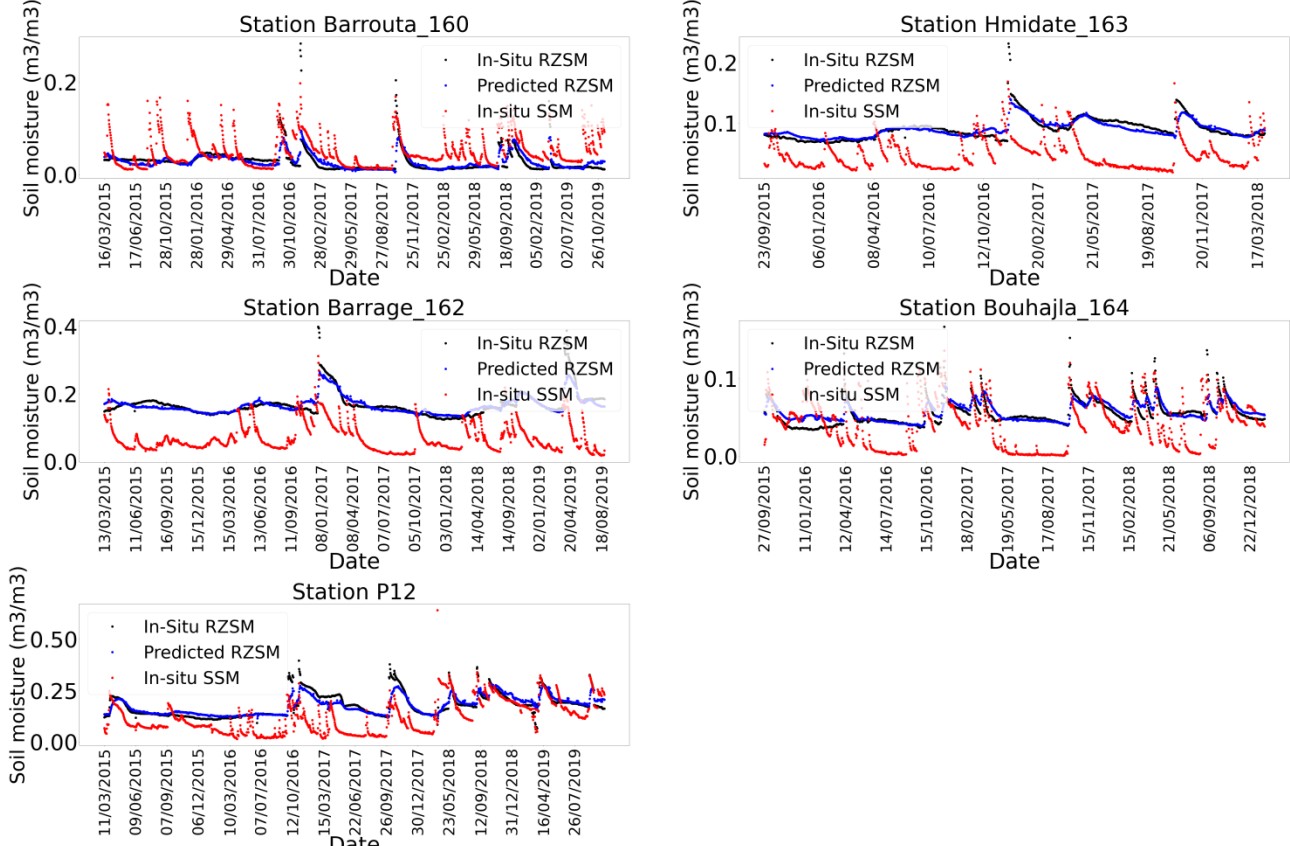

**Figure 4.** In situ SSM, in situ RZSM, and predicted RZSM series at the stations in the Kairouan Plain (Tunisia) with model
ANN_SSM_NDVI_EVAP-EFF-B60_EXP-FILT-T5 (cf. appendix G for larger figure format).
A second comparison can be conducted between the quality of fit of these independent datasets and training datasets.
Actually, the climate class of the Tunisian stations is 'Bsh' (cf. appendix A). At the training stage, no station falls into
the climate class 'Bsh' (cf. appendix A). However, some training stations fall under a similar climate class which is
'Bsk' (cf. appendix B). Table 5 presents correlation and RMSE values for these training stations and Tunisian sites with
both models ANN_SSM and ANN_SSM_NDVI_EVAP-EFF-B60_EXP-FILT-T5. For all training stations,
performance metrics are slightly enhanced with the most complex ANN model compared to reference model
ANN_SSM, except for stations GrouseCreek, Harmsway and Lind#1 which performance decreases. Overall, the range
of correlation values is similar for training and external validation stations with model ANN_SSM_NDVI_EVAP-EFF-
B60_EXP-FILT-T5 and RMSE is well reduced for Tunisian stations compared to training stations. Given the results on
unseen datasets, namely on Tunisia, the performance of the most complex ANN model is good as it is able to generalize
the patterns present in the training dataset.
**Table 5.** Performance metrics of models ANN_SSM and ANN_SSM_NDVI_EVAP-EFF-B60_EXP-FILT-T5 at training stations of
climate "Bsk" and Tunisian stations of climate "Bsh".

| Model | ANN_SSM | | ANN_SSM_NDVI_EVAP-EFF-B60_EXP-FILT-T5 | |
|---|---|---|---|---|
| | | Training stations (climate class 'Bsh') | | |
| Station | Correlation | RMSE | Correlation | RMSE |
| Banandra (OZNET) | 0.701 | 0.05 | 0.764 | 0.046 |

| | | | | |
|---|---|---|---|---|
| DRY-LAKE (OZNET) | 0.674 | 0.031 | 0.692 | 0.03 |
| CPER (SCAN) | 0.691 | 0.032 | 0.695 | 0.032 |
| EPHRAIM (SCAN) | 0.758 | 0.051 | 0.791 | 0.046 |
| GrouseGreek (SCAN) | 0.818 | 0.033 | 0.802 | 0.035 |
| HarmsWay (SCAN) | 0.705 | 0.034 | 0.622 | 0.038 |
| Lind#1 (SCAN) | 0.605 | 0.055 | 0.483 | 0.022 |
| External test stations (Tunisia) | | | | |
| Station | Correlation | RMSE | Correlation | RMSE |
| Barrouta_160 | 0.463 | 0.021 | 0.714 | 0.016 |
| Hmidate_163 | 0.318 | 0.019 | 0.834 | 0.011 |
| Barrage_162 | 0.416 | 0.035 | 0.864 | 0.019 |
| Bouhajla_164 | 0.435 | 0.016 | 0.733 | 0.01 |
| P12 | 0.581 | 0.047 | 0.861 | 0.029 |


At the South-Indian stations, the ANN_SSM model yielded a good agreement even without the integration of process-related features (Table 6). The NDVI added little to nonsignificant improvement at station Bheemanbidu. The same observation was made at the Italian site. The application of multiple features performed the best under arid conditions, e.g., in Tunisia. In the tropical and temperate climate regions, this was not the case. The presence of clouds in the MODIS NDVI and potential evapotranspiration products could explain this observation at sites of South-India and North-Italy. In South-India, for instance, the maximum variability in soil moisture occurred during the monsoon season, which is characterized by a large amount of clouds. Moreover, the coarse resolution of MODIS NDVI product makes it sometimes not adapted to the considered site. (Chen et al., 2016) investigated the impact of sample impurity and landscape heterogeneity on crop classification using coarse spatial resolution MODIS imagery. They showed that the sample impurity such as mixed crop types in a specific sample, compositional landscape heterogeneity that is the richness and evenness of land cover types in a landscape, and configurational heterogeneity that is the complexity of spatial structure of land cover types in a specific landscape are sources of uncertainty affecting crop area mapping when using coarse spatial resolution imagery. High resolution NDVI from sensors like Sentinel-2 could have been used in this exercise to mitigate the spatial resolution issue, however, MODIS data were privileged in order to provide NDVI and PET from the same sensor.

**Table 6.** Performance metrics of models ANN_SSM, ANN_SSM_NDVI and ANN_SSM_NDVI_EVAP-EFF-B60_EXP-FILT-T5 at the sites in South-India and Northern Italy.

| Model | ANN_SSM | | ANN_SSM_NDVI | | ANN_SSM_NDVI_EVAP-EFF_B60_EXP-FILT-T5 | |
|---|---|---|---|---|---|---|
| | | | INDIA | | | |
| Station | Correlation | RMSE | Correlation | RMSE | Correlation | RMSE |
| Madyanahundi | 0.813 | 0.04 | 0.78 | 0.042 | 0.744 | 0.044 |
| Bheemanbidu | 0.76 | 0.046 | 0.784 | 0.044 | 0.763 | 0.046 |
| Beechanalli2 | 0.825 | 0.038 | 0.787 | 0.04 | 0.743 | 0.044 |
| Beechanalli1 | 0.713 | 0.024 | 0.713 | 0.024 | 0.633 | 0.025 |
| | | | Italy | | | |
| Station | Correlation | RMSE | Correlation | RMSE | Correlation | RMSE |
| Landriano | 0.861 | 0.038 | 0.827 | 0.041 | 0.841 | 0.038 |


## 4 Discussion


Climate analysis of the results yielded by the different models indicated that among all models, the climate class with
the highest mean correlation change rate (Fig. 5) was class BWk (cf. appendix A), which regroups desert areas where
the link between SSM and RZSM is weak due to high evaporative rates. Class Dfa (cf. appendix A), which includes
areas experiencing harsh and cold winters, also yielded a high mean correlation change rate (>100%). Similarly, at
stations of this climate type, the link between the surface and root zone is poor. In regard to class Cfa (cf. appendix A),
in which more than 80% of the total stations belongs to SCAN network, the high mean correlation change rate could be
explained by the surface-subsurface decoupling phenomena detected within this network, as previously reported in
(Souissi et al., 2020). The model with the largest number of stations with improved predictions over the ANN_SSM
model predictions was ANN_SSM_NDVI_EVAP-EFF-B60_EXP-FILT-T5. Actually, the coupled use of process-
related features in the ANN models exerted a greater impact on the prediction accuracy than that exerted by the one-at-
a-time application of these features. In model ANN_SSM_NDVI_EVAP-EFF-B60_EXP-FILT-T5, the three process-
based features jointly employed seemed to counterbalance the weight of these three SSM features. In this model, the
process-related features were equally represented versus the SSM information depicted by these three features. The
redundancy of the considered SSM information could explain the limited impact of the one-at-a-time addition of
process-related features the joint addition of the three process-related features.
In addition, Karthikeyan and Mishra (2021) demonstrated that at root depths beyond 20 cm, the importance of SSM
was notably lower than that at the 20-cm depth, signifying decorrelation between surface and deeper SM values, which
is in accordance with the findings in (Souissi et al., 2020), and it was further revealed that vegetation exhibits a higher
importance than that of meteorological predictors LST and precipitation. Kornelsen and Coulibaly (2014) indicated
that evapotranspiration is the most important meteorological input for the prediction of soil moisture in the root zone
with the MLP, which reflects the importance of the water vapor flux in soil moisture state determination.

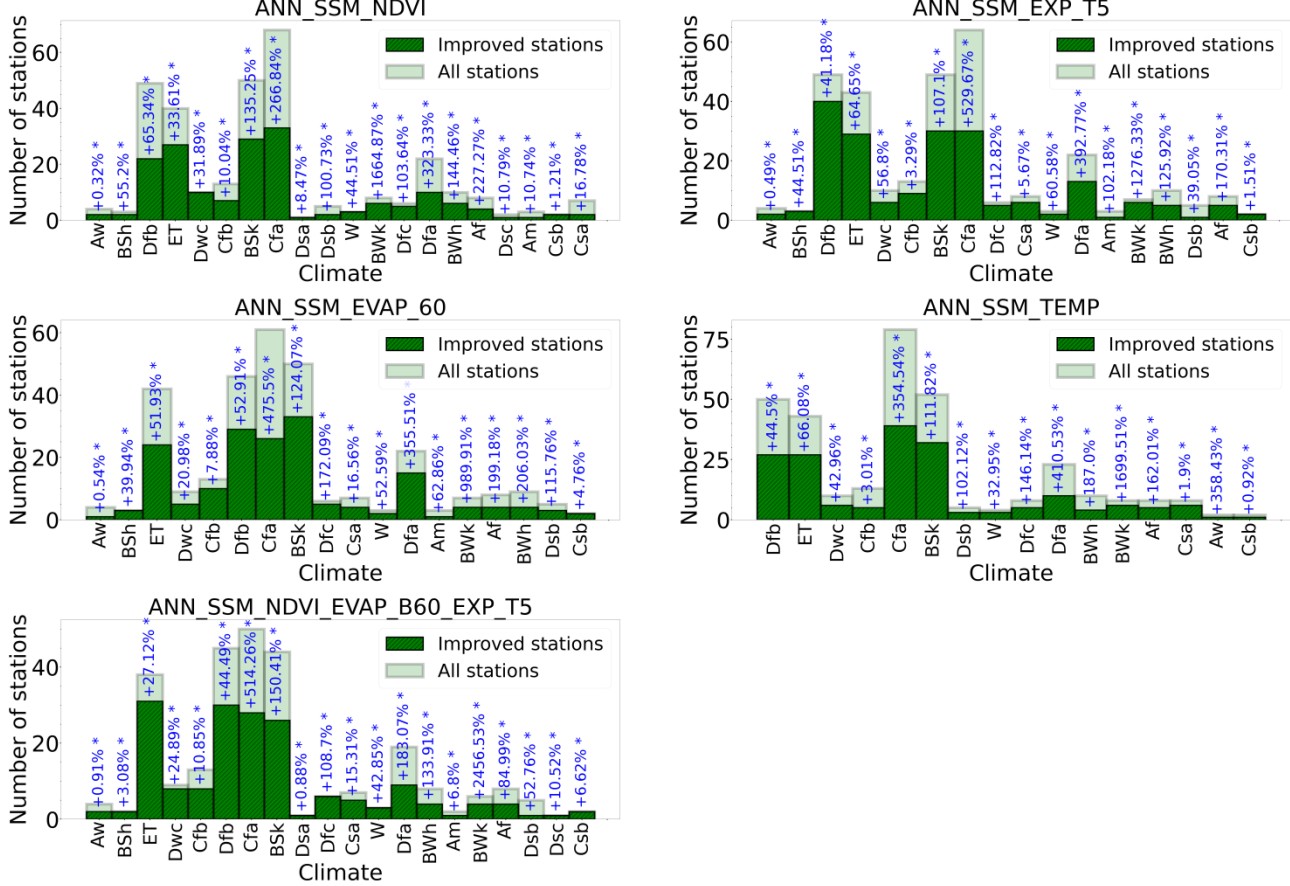

*Mean correlation change rate per climate class

**Figure 5.** Climate classification of the stations performing better with models (a) ANN_SSM_NDVI (b) ANN_SSM_EXP-T5 (c) ANN_SSM_EVAP-60 (d) ANN_SSM_TEMP and (e) ANN_SSM_NDVI_EVAP-EFF-B60_EXP-FILT-T5 compared to model ANN_SSM (Dark green corresponds to stations which correlation improved with complexified models, light green corresponds to total stations, rate in blue correspond to mean correlation change rate per climate class).

$$* corr\_change\_rate = mean(\frac{corr_{ANN\_SSM\_X} - corr_{ANN\_SSM}}{corr_{ANN\_SSM}} * 100) \ (5)$$

where X denotes a process-related variable (X ∈ ['NDVI', 'EXP-FILT-T5', 'EVAP-EFF-B60', 'TEMP']

The world map illustrated in Fig. 6, shows the best-performing ANN models based on the mean correlation change rate (Eq. 5). We assumed that the results in a given area of a specific climate class could be extended to other areas of the same climate class even if we did not consider the data for these areas. The climate classes without at least one station were marked in black and labeled with 'NO DATA'.

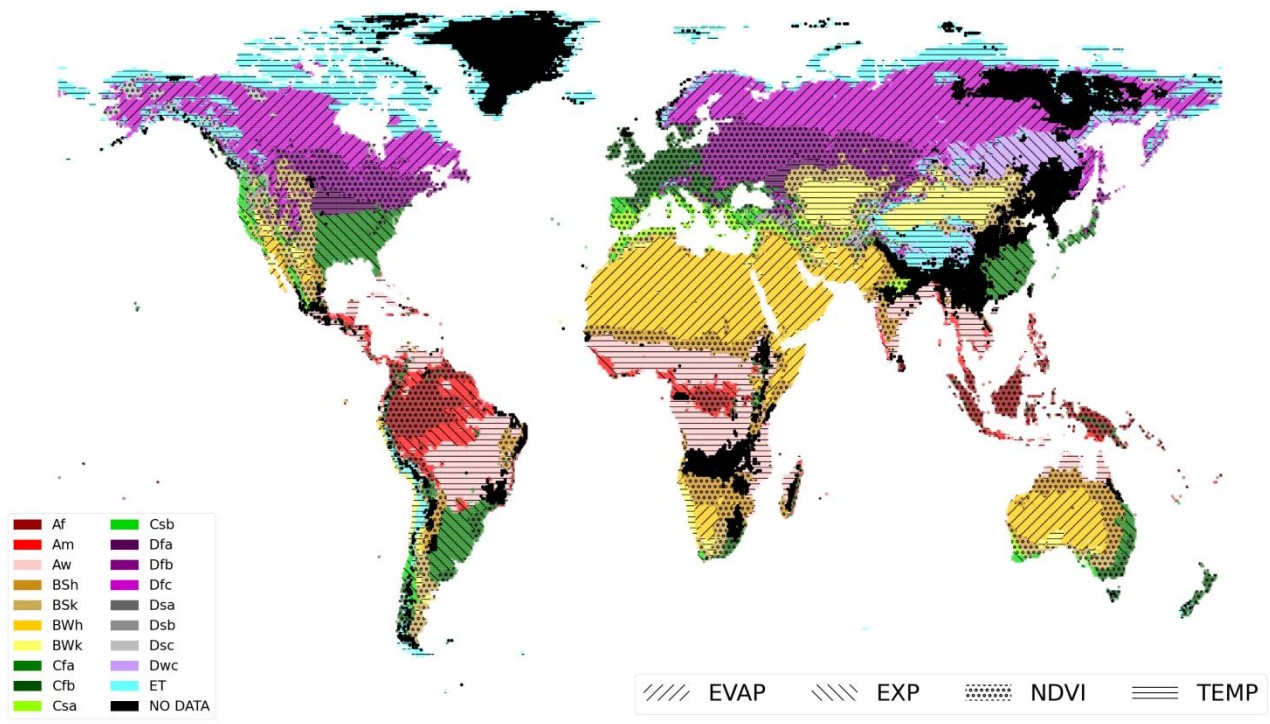

**Figure 6.** World map of best-performing ANN models per climate class based on the mean correlation change rate; colors correspond to climate classes (cf. appendix A), hatches correspond to the most contributive input to the predictions namely: EVAP (evaporation efficiency), EXP (exponential filter SWI), NDVI , TEMP (surface soil temperature).

In arid areas such as the eastern and western sides of the USA with high evaporation rates, ANN_SSM_EVAP-EFF-60 was the best performing model. Similarly, in bare areas of Africa, the Middle East and Australia where the Bwh climate class prevailed (arid desert hot climate; cf. appendix A), the evaporation efficiency was the best informative variable.

In the internal part of continental Europe and near the Mediterranean Basin, the NDVI was the most relevant indicator for RZSM estimation, where agricultural fields dominated. Similarly, the Great Plains region in the USA was deeply affected by the NDVI, as this region is a cultivated area. The same result could be obtained for regions belonging to climate class Bsh (arid steppe hot; cf. appendix A) and mainly covered by grassland and shrubland areas according to ESA CCI land cover maps.

In Nordic areas characterized by the ET climate class, the soil temperature was the most important root zone soil moisture indicator mainly because of the freeze–thaw events encountered in these regions. In tropical savannah wet areas (class Aw; cf. appendix A), the ANN_SSM_TEMP model was the best-performing model.

This classification definitely suffered limitations mainly provoked by the generalization of the climatic analysis results to areas not considered in this study. For instance, in regions of climate class Dfc (cold dry without a dry season, cold summer climate; cf. appendix A), we expect the temperature to serve as the most relevant indicator instead of the evaporation efficiency.

## 5    Conclusion

In this study, we developed several ANN models to estimate RZSM based either on solely in situ SSM information or on a group of process-related features, in addition to SSM, namely, the soil water index computed with a recursive

exponential filter, evaporation efficiency, NDVI and surface soil temperature. Different regions across the globe with distinct land cover and climate patterns were considered. The main conclusion of this study was that the consideration of more features in addition to SSM information could enhance the accuracy of RZSM predictions mainly in regions where the link between SSM and RZSM is weak.

In arid areas with high evaporation rates, the most informative feature was the evaporation efficiency. In regions with agricultural fields, the NDVI was, for example, the most relevant indicator to predict RZSM. Overall, the best performing model included the surface soil moisture, NDVI, SWI and evaporation efficiency as features. The robustness of the approach was further assessed through additional tests considering external sites in central Tunisia, India and Italy. Similarly, the process-related features exerted a positive impact on the prediction accuracy when combined with surface soil moisture in the case of Tunisia. The mean correlation across the five Tunisian stations sharply increased from 0.44 when only SSM was considered to 0.8 when all process-related features were combined with SSM. In India and Italy, the correlations were already high with the reference model ANN_SSM. The change in correlation after the addition of process-related features, namely NDVI, is about -0.04 which is nonsignificant, and is potentially because of the cloudy conditions in India and noisy MODIS products. Also the crop heterogeneity and sample impurity makes MODIS NDVI products not adapted to all sites.

As a research perspective, datasets can be separated in clusters corresponding to major climate classes and/or soil types. More analysis can be conducted in this direction to eventually make connections between the different inputs and climate/soil configurations.

Future work will examine the ability of the developed model to estimate RZSM across larger areas based on remote sensing global soil moisture products. The use of remote sensing derived soil moisture products may yield lower correlations with the reference model ANN_SSM which potentially implies further improvement when process-related features are added.

**Acknowledgments**

The PhD thesis of R. Souissi was financed by the ERANET RET-SIF project, and complementary financing was provided by the PRIMA Programme SMARTIES project. The authors thank the International Soil Moisture Network (ISMN) and supporting networks for providing the soil moisture data.

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

 **APPENDIX A**

**Climate classes (Köppen classification):**
- Af: Tropical Rainforest
- Am: Tropical Monsoon
- As: Tropical Savanna Dry
- Aw: Tropical Savanna Wet
- BWk: Arid Desert Cold
- BWh: Arid Desert Hot
- BWn: Arid Desert with Frequent Fog
- BSk: Arid Steppe Cold
- BSh: Arid Steppe Hot
- BSn: Arid Steppe with Frequent Fog
- Csa: Temperate Dry Hot Summer
- Csb: Temperate Dry Warm Summer
- Csc: Temperate Dry Cold Summer
- Cwa: Temperate Dry Winter, Hot Summer
- Cwb: Temperate Dry Winter, Warm Summer
- Cwc: Temperate Dry Winter, Cold Summer
- Cfa: Temperate without a Dry Season, Hot Summer
- Cfb: Temperate without a Dry Season, Warm Summer
- Cfc: Temperate without a Dry Season, Cold Summer
- Dsa: Cold Dry Summer, Hot Summer
- Dsb: Cold Dry Summer, Warm Summer
- Dsc: Cold Dry Summer, Cold Summer
- Dsd: Cold Dry Summer, Very Cold Winter
- Dwa: Cold Dry Winter, Hot Summer
- Dwb: Cold Dry Winter, Warm Summer
- Dwc: Cold Dry Winter, Cold Summer
- Dwd: Cold Dry Winter, Very Cold Winter
- Dfa: Cold Dry without a Dry Season, Hot Summer
- Dfb: Cold Dry without a Dry Season, Warm Summer
- Dfc: Cold Dry without a Dry Season, Cold Summer
- Dfd: Cold Dry without a Dry Season, Very Cold Winter
- ET: Polar Tundra
- EF: Polar Eternal Winter
- W: Water


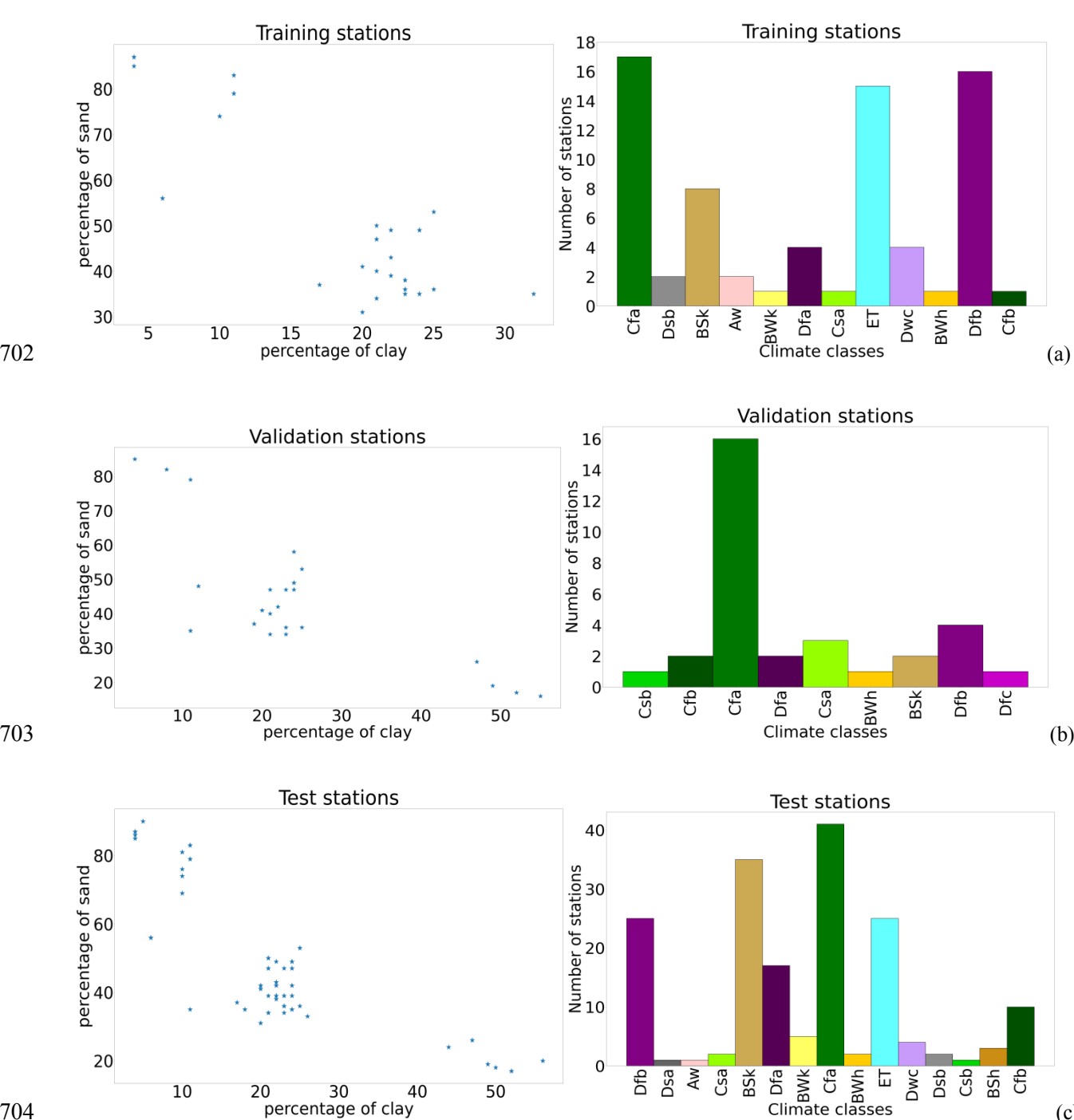


(a)
(b)
(c)
Figure B1. Climate and soil texture for (a) training stations (b) validation stations (c) test stations.




# APPENDIX C

Evaporation efficiency (section **2.2.2**):

The standard equations to compute evaporation efficiency ($\beta_3$) in (Merlin et al., 2010) are as follows:

$$\beta_3 = [\tfrac{1}{2} - \tfrac{1}{2}\cos(\pi\theta_L/\theta_{max})]^P \qquad for\ \theta_L \leq \theta_{max}\ (C3)$$

$$\beta_3 = 1\ for\ \theta_L > \theta_{max}$$

where:  - $\theta_L$ is the water content in the soil layer of thickness L.

- P is a parameter computed as follows:

$$P = (\tfrac{1}{2} + A_3\frac{L-L_1}{L_1})\frac{LE_p}{B_3}\ (C4)$$

- $\theta_{max}$ is the soil moisture at saturation.

- $LE_p$ is the potential evaporation.

- $L_1$ is the thinnest represented soil layer, and $A_3$ (unitless) and $B_3$ (W/m²) are the two best-fit parameters a priori depending on the soil texture and structure, respectively.

 **APPENDIX D**

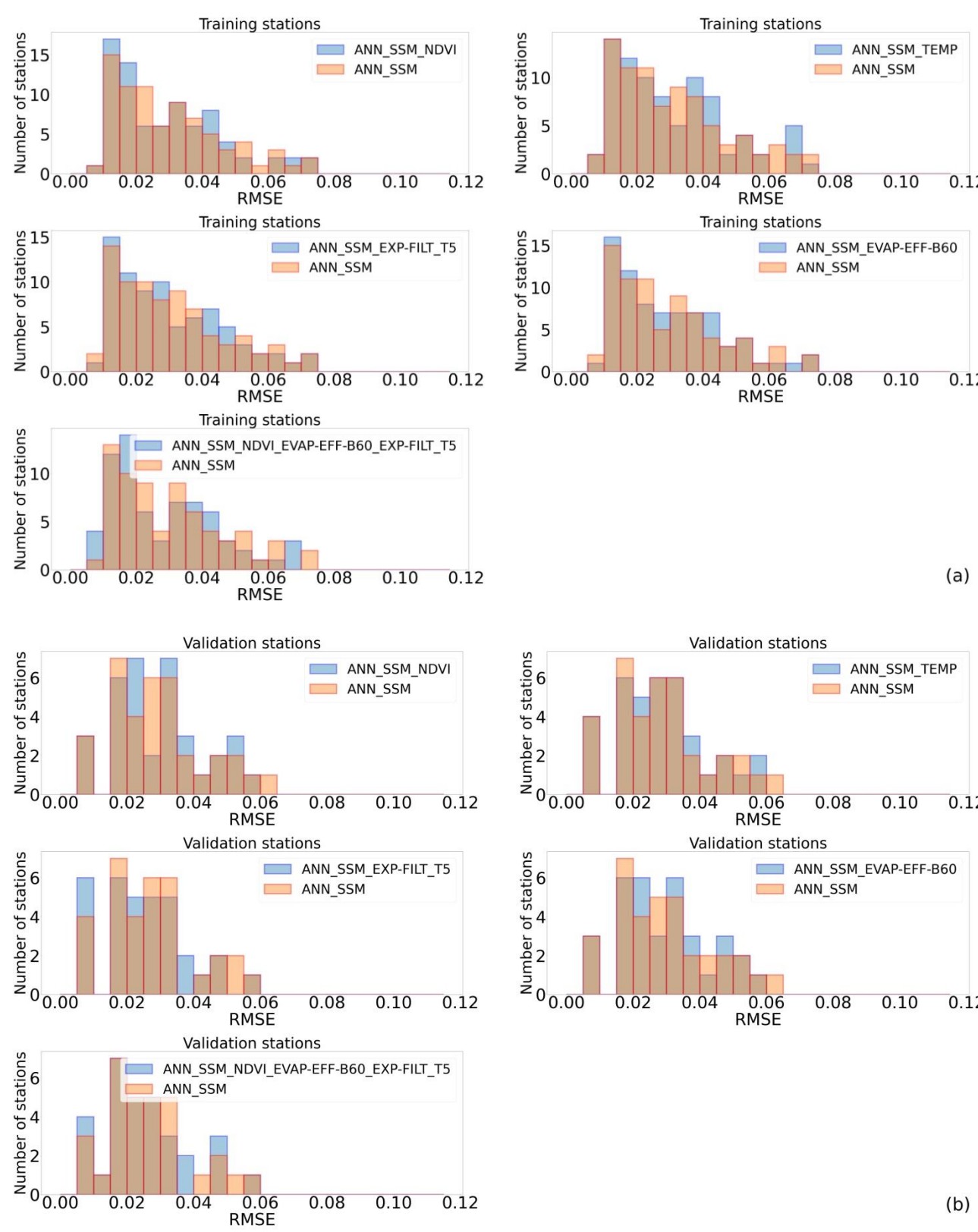

739

740

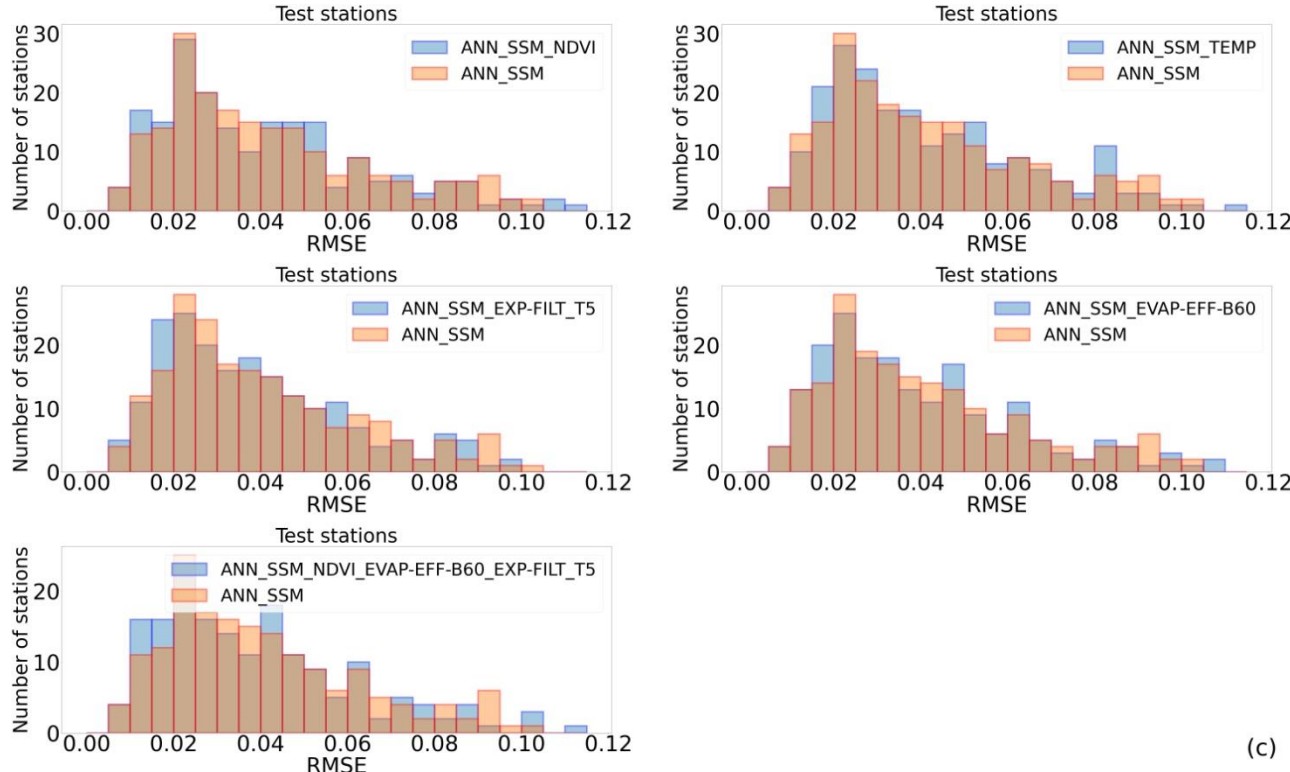

(c)

**Figure D1.** RMSE histograms of all tested ANN models compared to ANN_SSM (a) on training stations (b) on validation stations (c) on test stations.

**APPENDIX E**

Training stations:

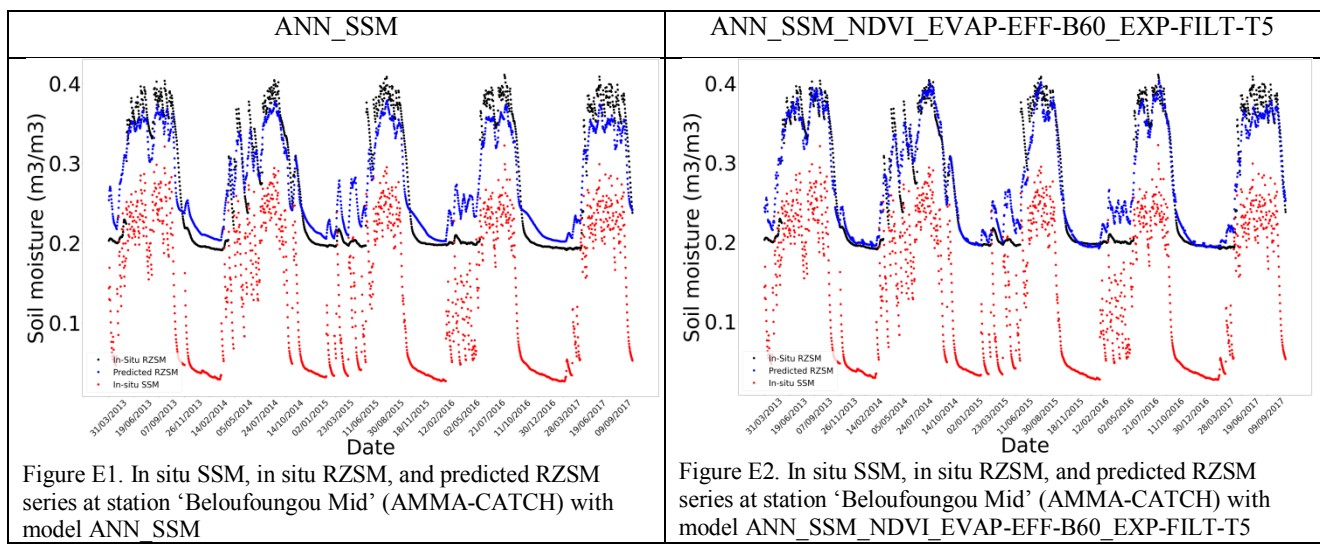

| ANN_SSM | ANN_SSM_NDVI_EVAP-EFF-B60_EXP-FILT-T5 |
|---|---|
| Figure E1. In situ SSM, in situ RZSM, and predicted RZSM series at station 'Beloufoungou Mid' (AMMA-CATCH) with model ANN_SSM | Figure E2. In situ SSM, in situ RZSM, and predicted RZSM series at station 'Beloufoungou Mid' (AMMA-CATCH) with model ANN_SSM_NDVI_EVAP-EFF-B60_EXP-FILT-T5 |


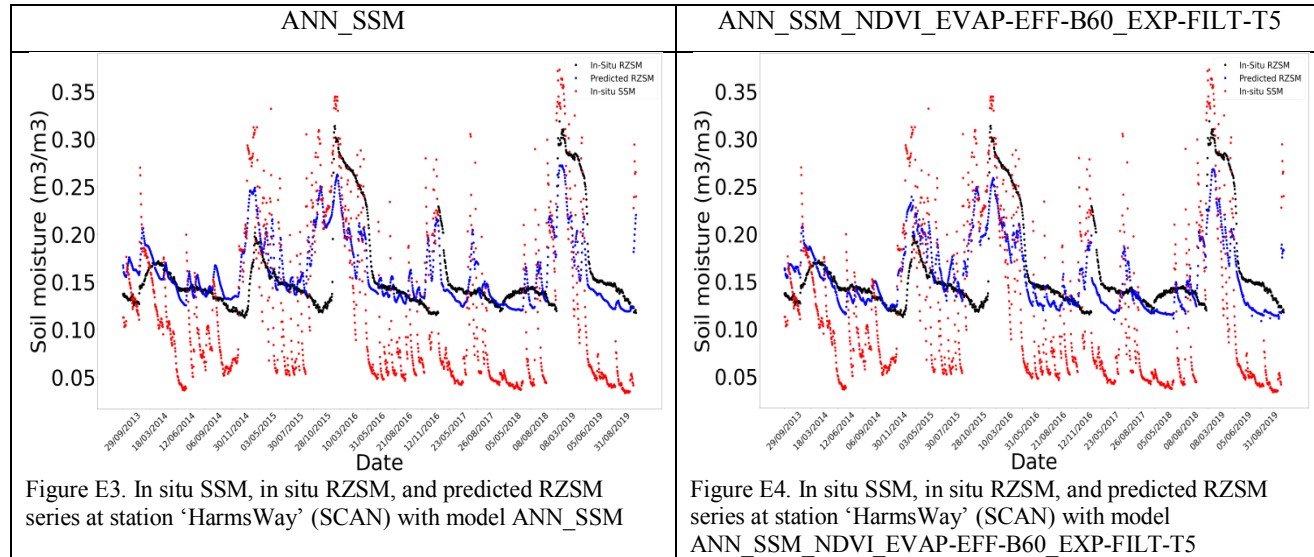

| ANN_SSM | ANN_SSM_NDVI_EVAP-EFF-B60_EXP-FILT-T5 |
|---|---|
| Figure E3. In situ SSM, in situ RZSM, and predicted RZSM series at station 'HarmsWay' (SCAN) with model ANN_SSM | Figure E4. In situ SSM, in situ RZSM, and predicted RZSM series at station 'HarmsWay' (SCAN) with model ANN_SSM_NDVI_EVAP-EFF-B60_EXP-FILT-T5 |


Validation stations

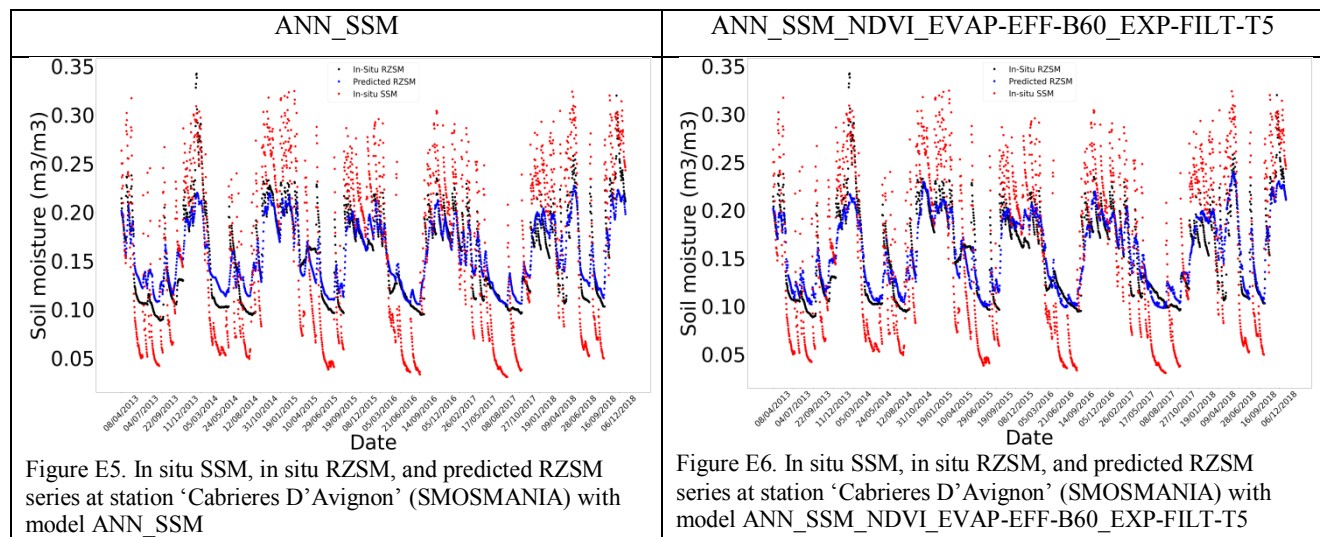

| ANN_SSM | ANN_SSM_NDVI_EVAP-EFF-B60_EXP-FILT-T5 |
|---|---|
| Figure E5. In situ SSM, in situ RZSM, and predicted RZSM series at station 'Cabrieres D'Avignon' (SMOSMANIA) with model ANN_SSM | Figure E6. In situ SSM, in situ RZSM, and predicted RZSM series at station 'Cabrieres D'Avignon' (SMOSMANIA) with model ANN_SSM_NDVI_EVAP-EFF-B60_EXP-FILT-T5 |


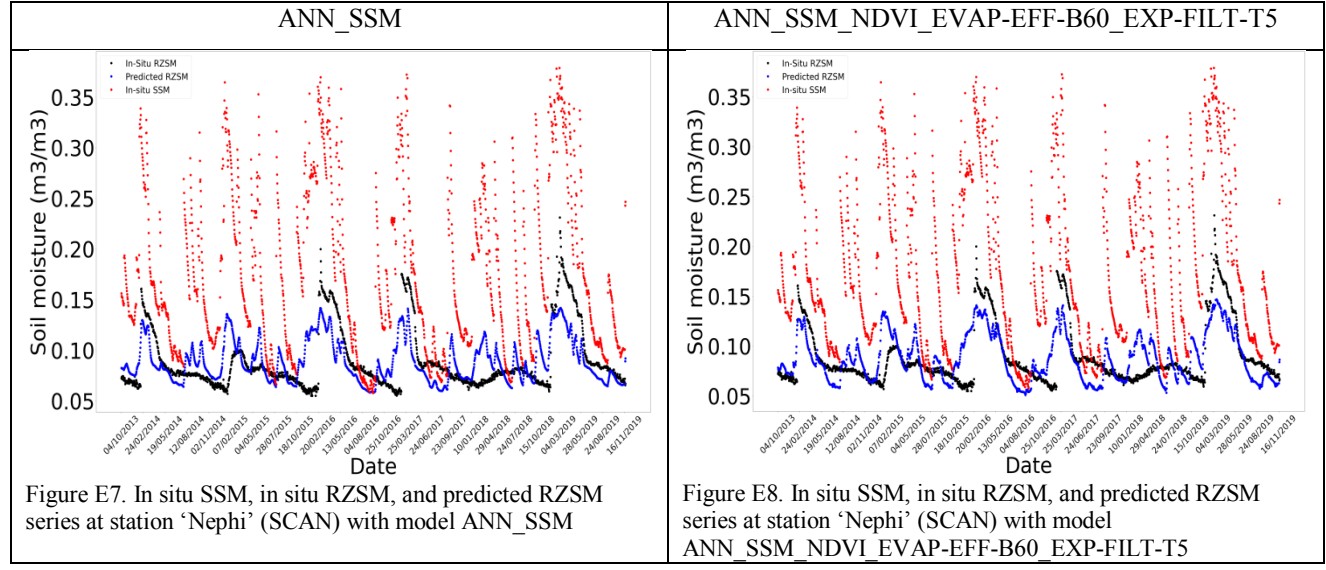

| ANN_SSM | ANN_SSM_NDVI_EVAP-EFF-B60_EXP-FILT-T5 |
|---|---|
| Figure E7. In situ SSM, in situ RZSM, and predicted RZSM series at station 'Nephi' (SCAN) with model ANN_SSM | Figure E8. In situ SSM, in situ RZSM, and predicted RZSM series at station 'Nephi' (SCAN) with model ANN_SSM_NDVI_EVAP-EFF-B60_EXP-FILT-T5 |


ISMN test stations

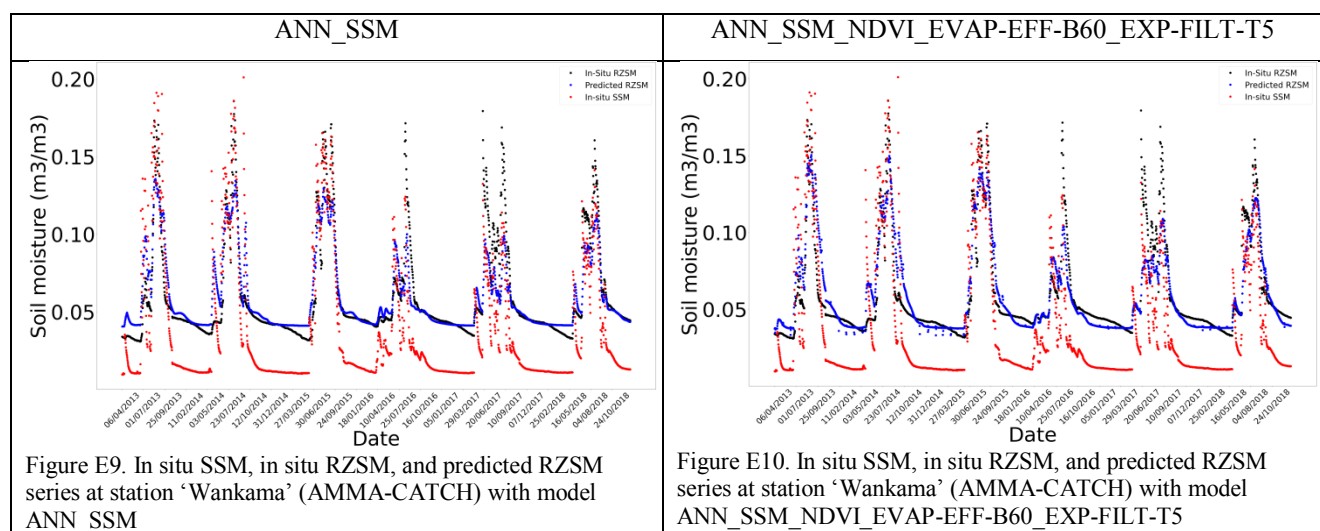

| ANN_SSM | ANN_SSM_NDVI_EVAP-EFF-B60_EXP-FILT-T5 |
|---|---|
| Figure E9. In situ SSM, in situ RZSM, and predicted RZSM series at station 'Wankama' (AMMA-CATCH) with model ANN_SSM | Figure E10. In situ SSM, in situ RZSM, and predicted RZSM series at station 'Wankama' (AMMA-CATCH) with model ANN_SSM_NDVI_EVAP-EFF-B60_EXP-FILT-T5 |


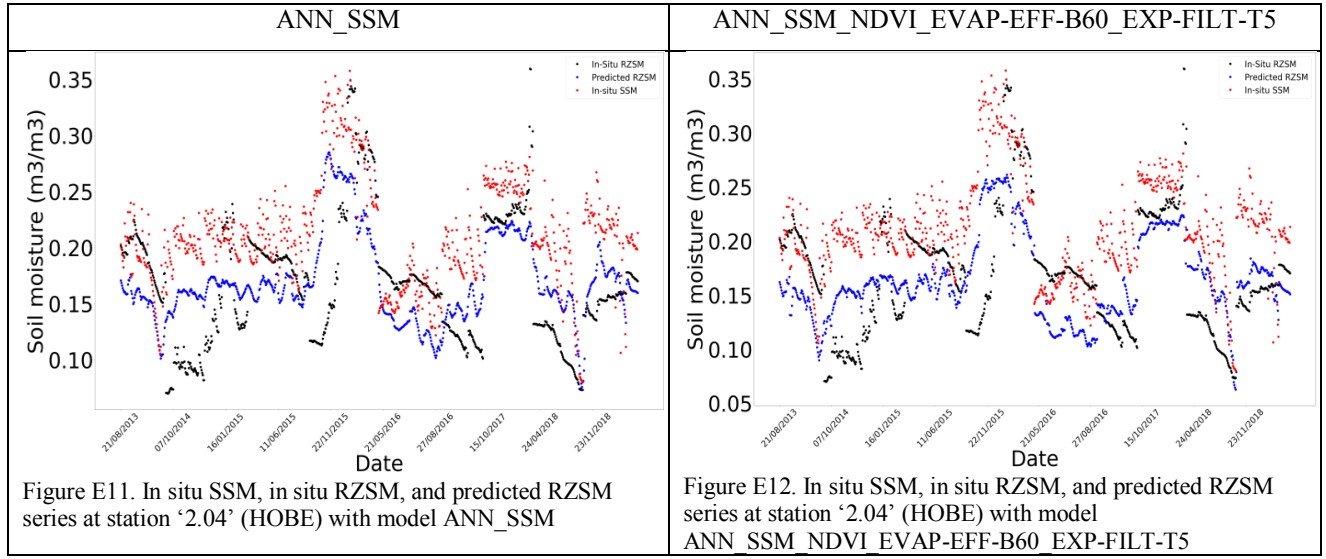

| ANN_SSM | ANN_SSM_NDVI_EVAP-EFF-B60_EXP-FILT-T5 |
|---|---|
| Figure E11. In situ SSM, in situ RZSM, and predicted RZSM series at station '2.04' (HOBE) with model ANN_SSM | Figure E12. In situ SSM, in situ RZSM, and predicted RZSM series at station '2.04' (HOBE) with model ANN_SSM_NDVI_EVAP-EFF-B60_EXP-FILT-T5 |


**APPENDIX F**

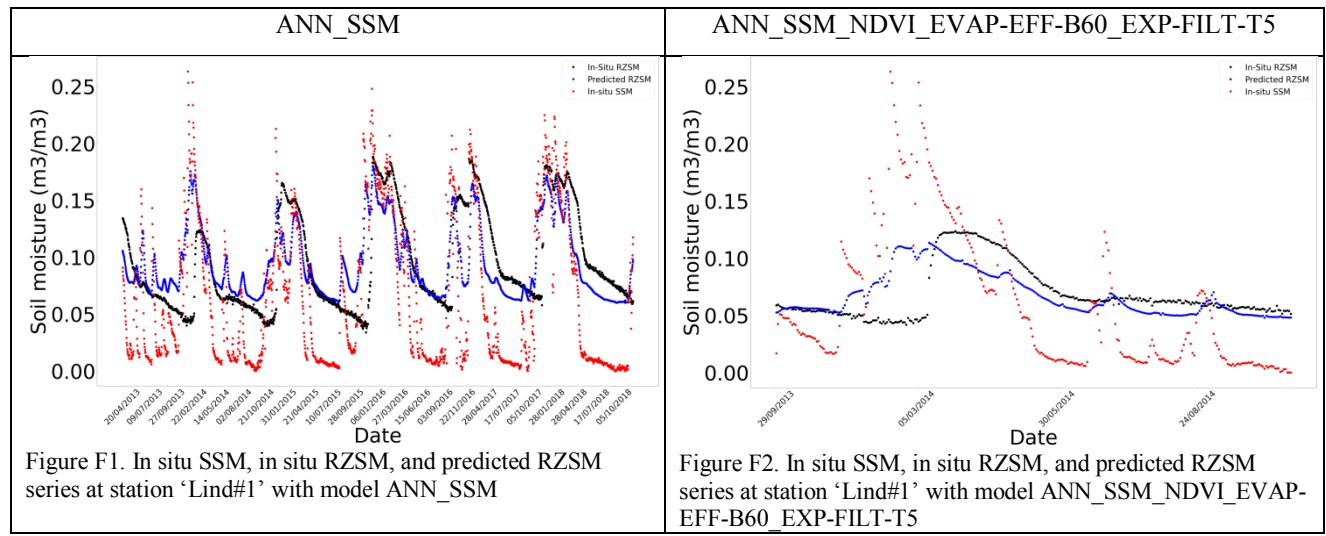

Figure F1. In situ SSM, in situ RZSM, and predicted RZSM series at station 'Lind#1' with model ANN_SSM

Figure F2. In situ SSM, in situ RZSM, and predicted RZSM series at station 'Lind#1' with model ANN_SSM_NDVI_EVAP-EFF-B60_EXP-FILT-T5

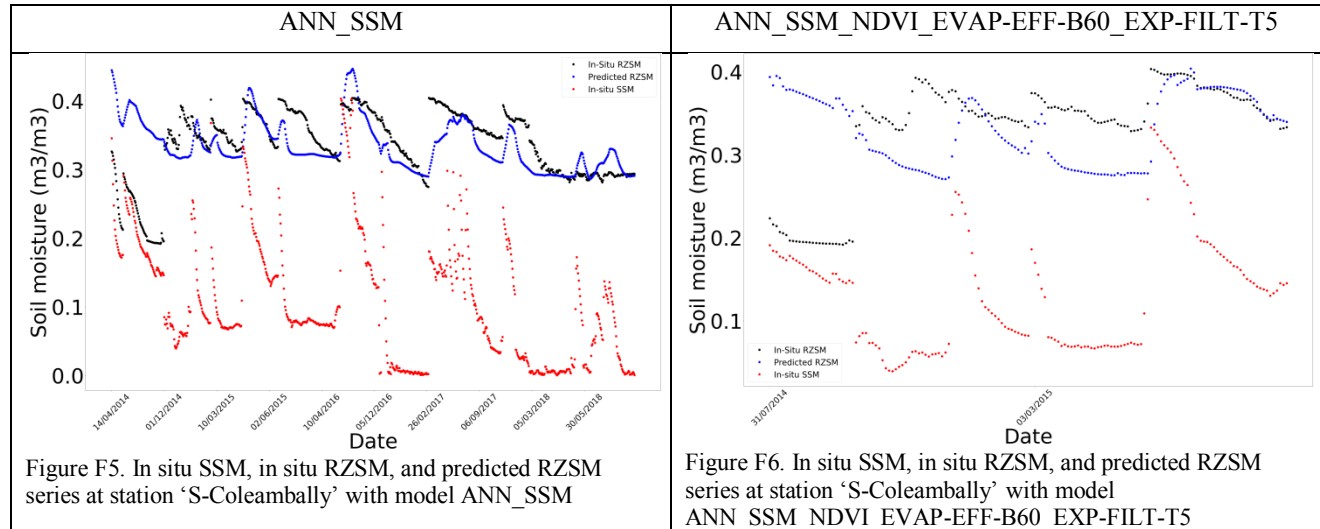

Figure F3. In situ SSM, in situ RZSM, and predicted RZSM series at station 'PineNut' with model ANN_SSM

Figure F4. In situ SSM, in situ RZSM, and predicted RZSM series at station 'PineNut' with model ANN_SSM_NDVI_EVAP-EFF-B60_EXP-FILT-T5

Figure F5. In situ SSM, in situ RZSM, and predicted RZSM series at station 'S-Coleambally' with model ANN_SSM

Figure F6. In situ SSM, in situ RZSM, and predicted RZSM series at station 'S-Coleambally' with model ANN_SSM_NDVI_EVAP-EFF-B60_EXP-FILT-T5

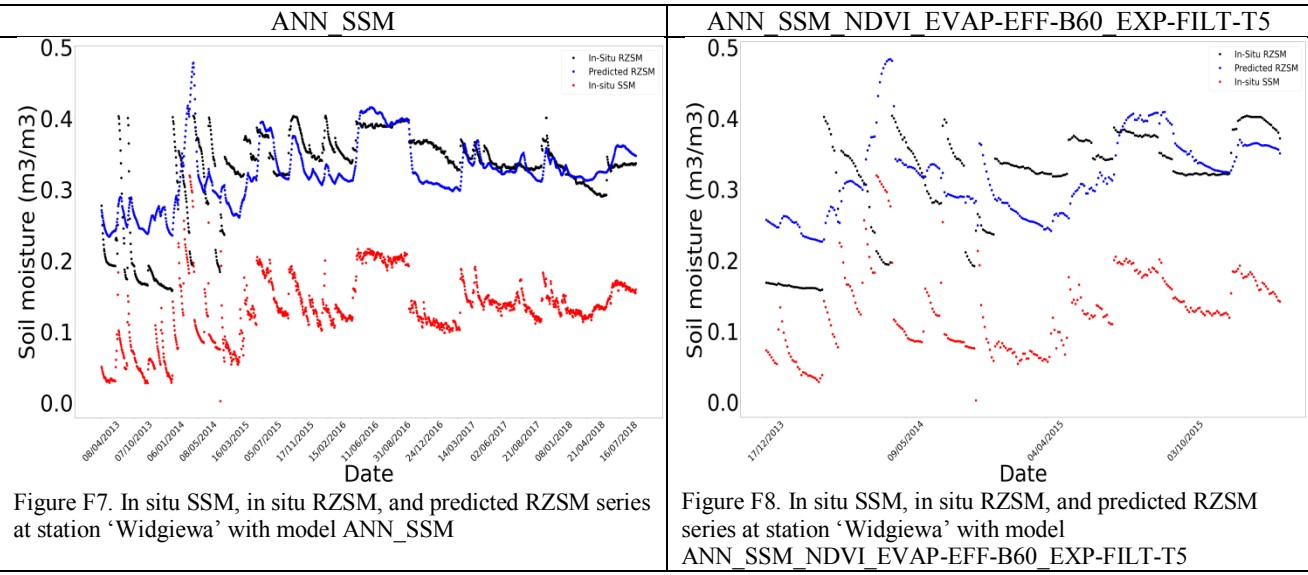

Figure F7. In situ SSM, in situ RZSM, and predicted RZSM series at station 'Widgiewa' with model ANN_SSM

Figure F8. In situ SSM, in situ RZSM, and predicted RZSM series at station 'Widgiewa' with model ANN_SSM_NDVI_EVAP-EFF-B60_EXP-FILT-T5

















**APPENDIX G**

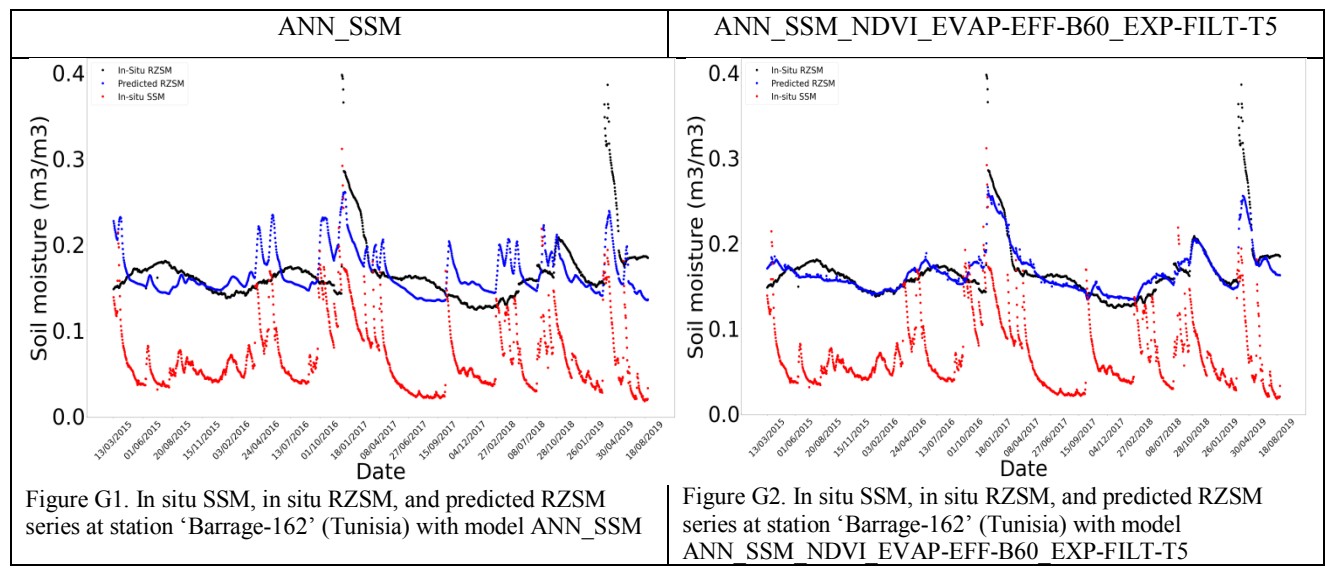

Figure G1. In situ SSM, in situ RZSM, and predicted RZSM series at station 'Barrage-162' (Tunisia) with model ANN_SSM

Figure G2. In situ SSM, in situ RZSM, and predicted RZSM series at station 'Barrage-162' (Tunisia) with model ANN_SSM_NDVI_EVAP-EFF-B60_EXP-FILT-T5

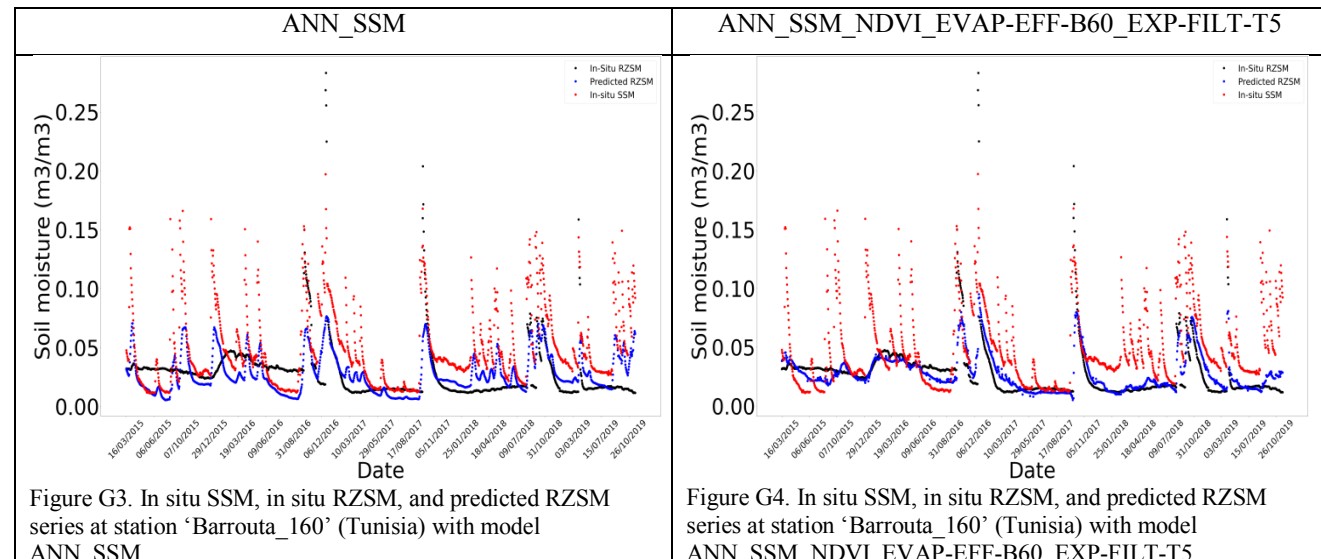

Figure G3. In situ SSM, in situ RZSM, and predicted RZSM series at station 'Barrouta_160' (Tunisia) with model ANN_SSM

Figure G4. In situ SSM, in situ RZSM, and predicted RZSM series at station 'Barrouta_160' (Tunisia) with model ANN_SSM_NDVI_EVAP-EFF-B60_EXP-FILT-T5

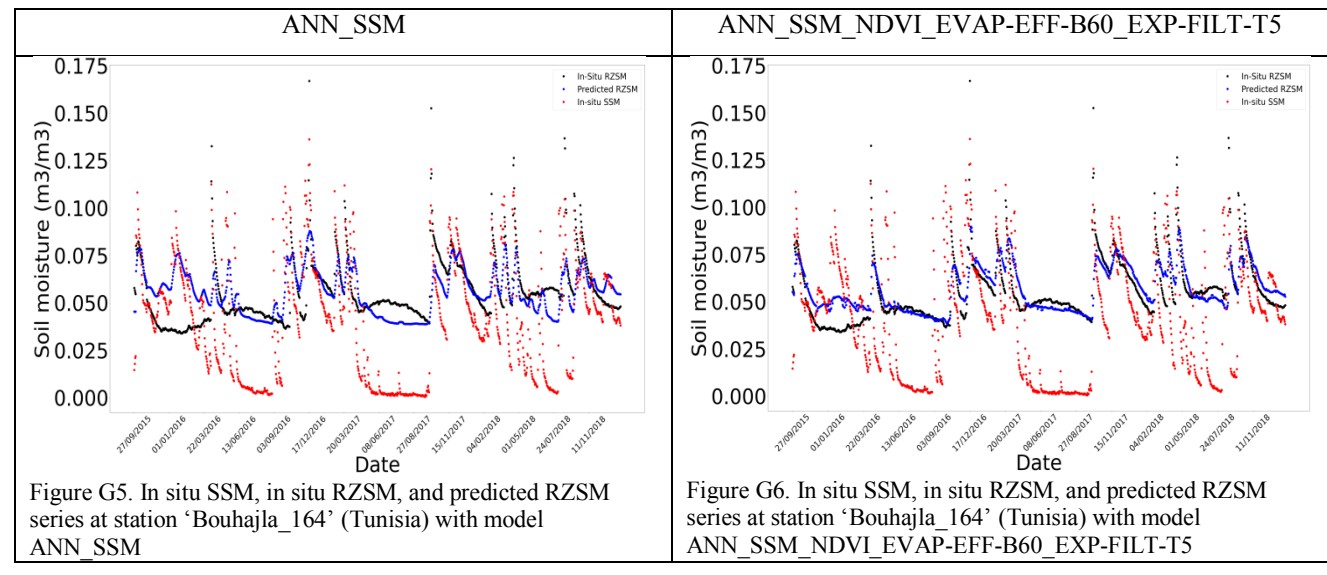

Figure G5. In situ SSM, in situ RZSM, and predicted RZSM series at station 'Bouhajla_164' (Tunisia) with model ANN_SSM

Figure G6. In situ SSM, in situ RZSM, and predicted RZSM series at station 'Bouhajla_164' (Tunisia) with model ANN_SSM_NDVI_EVAP-EFF-B60_EXP-FILT-T5

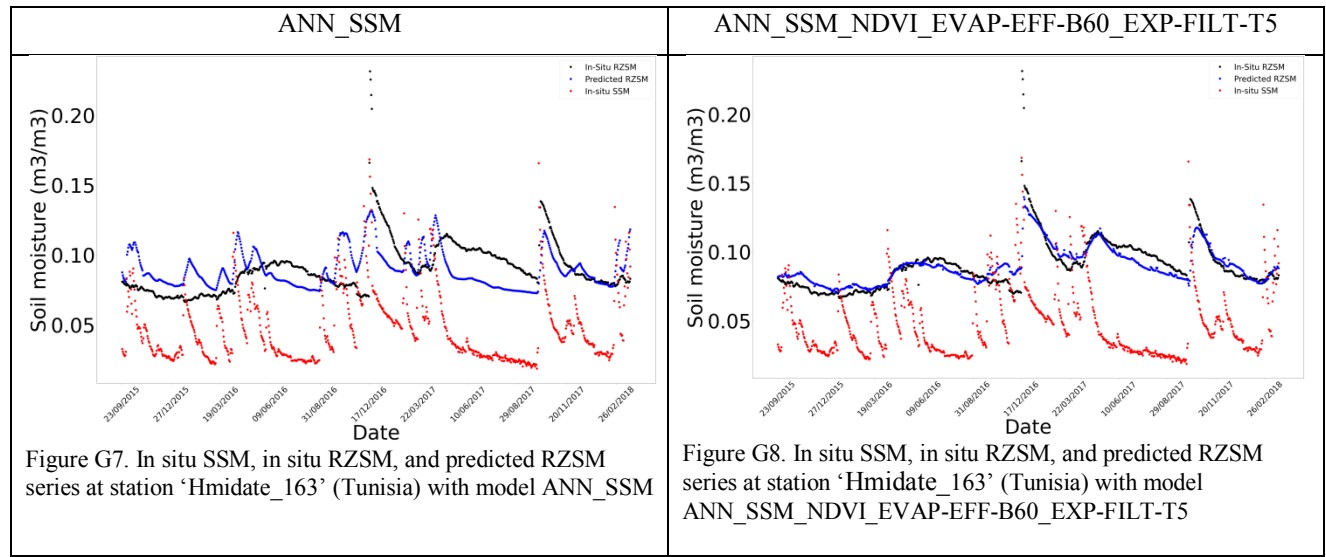

| ANN_SSM | ANN_SSM_NDVI_EVAP-EFF-B60_EXP-FILT-T5 |
|---|---|
| Figure G7. In situ SSM, in situ RZSM, and predicted RZSM series at station 'Hmidate_163' (Tunisia) with model ANN_SSM | Figure G8. In situ SSM, in situ RZSM, and predicted RZSM series at station 'Hmidate_163' (Tunisia) with model ANN_SSM_NDVI_EVAP-EFF-B60_EXP-FILT-T5 |

794

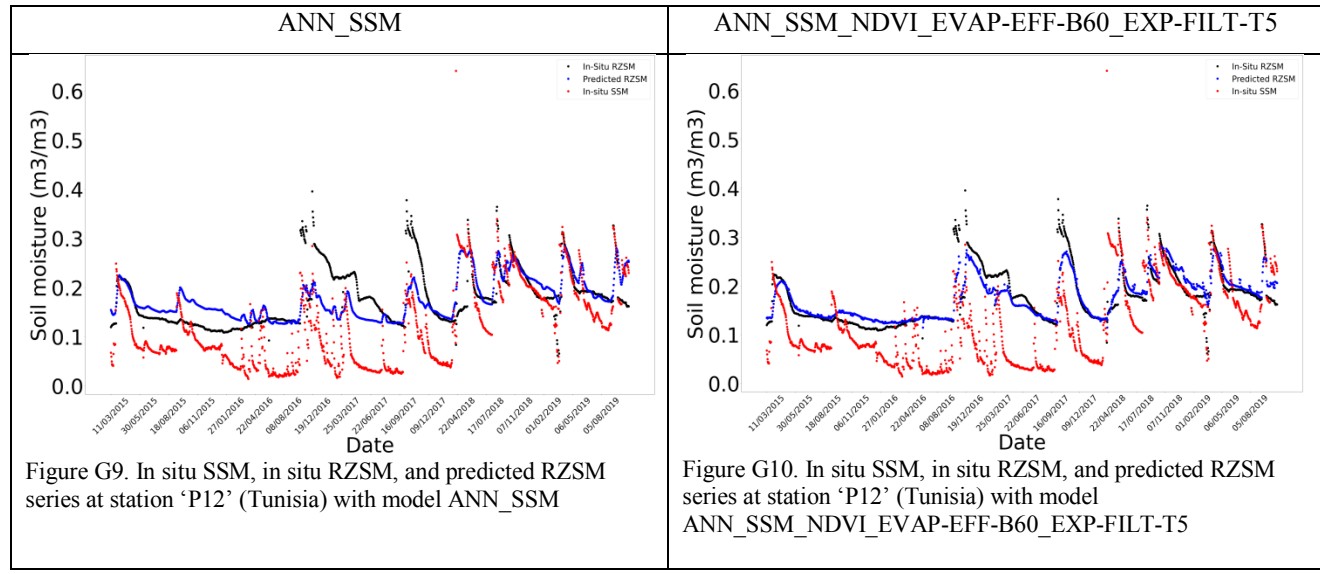

| ANN_SSM | ANN_SSM_NDVI_EVAP-EFF-B60_EXP-FILT-T5 |
|---|---|
| Figure G9. In situ SSM, in situ RZSM, and predicted RZSM series at station 'P12' (Tunisia) with model ANN_SSM | Figure G10. In situ SSM, in situ RZSM, and predicted RZSM series at station 'P12' (Tunisia) with model ANN_SSM_NDVI_EVAP-EFF-B60_EXP-FILT-T5 |

795
796