# Peer review of "Integrating process-related information into an ANN for root-zone soil moisture prediction"

_Hydrology and Earth System Sciences, 2022_

## Author Comment (AC1)

**Response to comments of Referee #1**

Please find in Black the reviewer's comments and in blue our answers.

**General comments:**

"In this manuscript, the authors estimate soil water content in the root zone (RZSM) using data-driven artificial neural networks (ANNs). While a previous study used soil surface moisture (SSM) data as input of the ANNs, the originality of this study is to test complementary input variables such as the normalized difference vegetation index (NDVI), soil temperature, and mathematical transformations of the SSM and potential evapotranspiration (PET), called soil water index (SWI) and evaporative efficiency, respectively.

The results indicate that RZSM predicted with the ANNs using the complementary input data tend to be more correlated to measured RZSM than predictions using temporal integrations of SSM data alone. The ANNs were trained and validated using a large amount of observations throughout stations around the world, and the robustness of the predictions was tested against RZSM data in independent stations.

This study addresses an important topic relevant to the readership of HESS. A method proving high-throughput estimations of RZSM at large scale from satellite images could become a game-changer for global water circulation modelling and crop modelling. Overall, the figures and structure of the paper need improvement. Substantial effort would also be needed to improve the presentation of the results and develop their discussion in light of the broader literature.

I am a bit concerned that I could not find a comparison of the qualities of fit of the training and validation datasets, so that in their current form, the results cannot exclude the possibility that improvements of the quality of fit with more complex models are due to overfitting allowed by their higher numbers of degrees of freedom. I am also disappointed that time series of SSM and RZSM from only five stations at the same site are shown, which contrasts with the wealth of data used in the study. There is extensive room to display the wide range of examples of quality of fit (from poor to excellent) across stations and ANN types in the Results section and appendices. Please also consider providing scripts and data as appendices upon publication, as this has become a widespread Open Science practice in major journals."

**Reply:** We thank the reviewer for the constructive comments that guided us to improve the paper. The suggested modifications impact the paper in all the sections notably: clarifications about the RZSM, clarifications about the evaporation efficiency, detailed results of training, validation and test experiments, time series of good and

less good quality of fit, comparisons of the quality of fit between independent data and training data and further discussion about the limitations and perspectives.

Also, the structure of the revised manuscript has been modified notably by: moving text in the appropriate sections, regenerating figures and enriching appendices.

Please find below, point-by-point detailed responses to the specific comments.

**\*Specific comments:**

**Comment: "**Title: I really like the idea to use process-based models' outputs as inputs of ANNs. It seems to me that the complementary data (PET, NDVI, temperature, …) used as novel inputs (directly or indirectly) to the ANNs could be inputs of process-based models meant to predict RZSM. However, I am less comfortable with labelling variables as "process-based" (see e.g. line 88-89), as I think such a label may characterize a model but not individual variables. Some of these variables are used to calculate indices (SWI and evaporation efficiency) related to soil water dynamics, but I have doubts that the equations behind these indices actually describe processes,though I agree that they are related to processes, just like the variables. Thus, using the term "process-based information" in the title seems misleading to me. An expression like "key hydrological indices" would probably be more representative of the content of the study."

**Reply:** We replaced the « process-based » by « process-related » in all the text, including the title, to avoid any confusion.

We can justify the fact that the variables are outputs of physical processes models. First for the Soil Water Index (SWI) computed based on the time characteristic parameter T,  we can cite the following paragraph in (Albergel et al. 2008) : «*In this case, the parameter T represents a characteristic time length. This parameter can be considered as a surrogate parameter for all the processes affecting the temporal dynamics of soil moisture, such as the thickness of the soil layer, soil hydraulic properties, evaporation, run-off and vertical gradient of soil properties (texture, density). T represents the time scale of soil moisture variation, in units of day (Ceballos et al., 2005). Different important processes such as transpiration are not considered in Eq. (2). Additionally, it is assumed that the soil hydraulic conductivity is constant while it may vary in reality by several orders of magnitude depending on the soil moisture conditions (Wagner et al., 1999).*»

As for evaporation efficiency, Merlin et al. (2010) describes the approach they developed as follows: «*It is found that (i) soil evaporative efficiency cannot be considered as a function of soil moisture only because it also depends on potential evaporation, (ii) retention forces in the soil increase in reaction to an increase of potential evaporation, and (iii) the model is able to accurately predict the soil evaporation process for soil layers with an arbitrary thickness up to 100 cm.*»

**Comment: "**Introduction

Line 36 (L36):  Here I got confused about the meaning of RZSM, and only understood pages later that it meant a *point* observation of soil water content in the root zone, not the integral, or average of soil water content from 30 to 100 cm depth. I think that given its central role, it is very important that the authors clearly define this variable early on in the manuscript to avoid confusion It is also unclear why say 1m as I could not find observation points reaching 1m depth in the document."

**Reply:** This point has been further clarified in the introduction part as follows : «*This essential climate variable (ECV) consists of two components, namely, surface soil moisture (SSM) (0–5 cm) and root-zone soil moisture (RZSM). RZSM corresponds to the soil moisture in the region in which the main vegetation rooting network is developing. Its definition varies depending on vegetation type and pedoclimatic conditions.*»

Further in the revised introduction, we have added this clarification: «*In this work, RZSM refers to a point observation of water content in a depth ranging between 30 and 55cm.*»

**Comment: "**Material and Methods

Figure 1: The fact that all input arrows get into the same "scaling" circle before the ANN I found a bit confusing, as the terms "scaling" and "descaling" are not defined, and it seems like all inputs get into the same neuron. I think the graphical representation could be improved, and space could really be used to provide clarifications in the caption and in the text (the current caption is only 5 words…) . The term "pheno." which I guess indicates the phenological stage is also not defined. At some point the NDVI is connected to the process of "growth" in the text. Probably worth selecting a single term to avoid confusion. Note that as discussed earlier, I would consider that the variable NDVI cannot be called "process-based", and that the process of growth, or the phenological stage change, is not modelled here. It is also worth mentioning in the caption that all the tested ANNs are variations of the one represented in Fig. 1, with all inputs or some of the inputs removed."

**Reply:** We agree with the reviewer. In the revised version of the paper, we have explained the "scaling" and "descaling" blocks before figure 1 as follows: «*Standard scaling is applied to each dataset separately so that the different inputs fall into the same range of values, then the ANN outputs are descaled to make the comparison with actual values of RZSM possible.*»

Also, the caption of figure 1 has been modified. It now reads:  «*Overview of the processing configuration showing the components of the model: the tested models are variations of this ANN with a different combination of inputs (see Table 1). The scaling and descaling  are applied to each dataset separately.*»

[Figure]

In the revised paper, the term "phen." (phenology) in figure 1 has also been replaced by plant growth which we agree is more relevant. Also, NDVI is described as an index to infer vegetation growth. We agree that the use of NDVI time series is not linked to any modeling exercise and it cannot be considered as process-based variable. In the submitted manuscript, the NDVI related sentence reads: *"We considered the infiltration process through the soil water index (SWI) computed with a recursive exponential filter and the evaporation process through the evaporative efficiency computed based on a MODIS remote sensing dataset and simplified analytical model, while vegetation growth was expressed through normalized difference vegetation index (NDVI) time series"*

To emphasize the contrast between NDVI and other variables (SWI and evaporation efficiency), the new sentence now reads: «*We considered the infiltration process through the soil water index (SWI) computed with a recursive exponential filter and the evaporation process through the evaporative efficiency computed based on a MODIS remote sensing dataset and simplified analytical model, while vegetation growth was not modeled and only inferred through normalized difference vegetation index (NDVI) time series.*»

**Comment:** "L100: "three SSM features" are mentioned here but I couldn't find a clear definition or equation for them in the manuscript (only a brief mention that they are rolling averages of SSM in the caption of table 1, then at line 244, which comes too late and without enough details)."

**Reply:** The three SSM features are respectively the backward rolling average values of hourly in-situ SSM over 10, 30, and 90 days. We agree with the reviewer that this definition comes too late in the text. In the revised paper, this definition has been added earlier in the text as follows : *«Each model has one or more process-related features in addition to three SSM features which correspond to backward rolling averages of in-situ SSM computed over 10,30 and 90 days. All the ANN model hyperparameters remain the same except the number of input features.»*

**Comment:** "L114: It is confusing that the authors mention "a root zone depth varying between 30 and 60 cm" as "root zone depth" misleads the reader into believing that the bottom of the root zone is located between 30 and 60 cm. Please clarify that "the root zone soil moisture observation point is located between 30 and 60 cm". Also, it is unclear why 60 cm is mentioned, while the deepest observation point is 55 cm."

**Reply:** We agree with the reviewer. The shallowest depth we considered is equal to 30cm as shown in Table 2. The deepest RZSM acquisition point is located at 55cm. The text has been modified accordingly as follows: *«the RZSM observation point is located between 30 and 55cm.»*

The soil moisture probes, even though they are point probes, are representative of a certain thickness of soil (several centimeters). Also, they are not installed totally horizontally which influences the depth. Finally, at depths above 50 cm the variation of the RZSM is not very significant (highly correlated in depth) except for fast capillary rise flood events.

**Comment:** "Figure 2: Words and points are too small to read."

**Reply:** Figure 2 has been regenerated with larger points and fontsize.

[Figure]

*Figure 2. International Soil Moisture Network (ISMN) network distribution (adapted from the ISMN web data portal (https://www.geo.tuwien.ac.at/insitu/data_viewer/); scale: 1 cm=1000 km).*

**Comment:** "L124-147: Please improve the descriptions for the sites to make them easier to compare. In one case, coordinates are mentioned, in another one yearly rainfall and PET, while the specific soil type at the observation site is only mentioned for one site, etc. Also, ideally a few descriptors (e.g. climate and soil type) for the other sites used in the training and validation steps should be provided in appendix, not necessarily individual descriptors, but at least "population" descriptors giving an idea of the frequency of specific soil types across sites. Climates are mentioned with acronyms that are not defined. They can be found in appendices, but the reference to the appendices is missing."

**Reply:** We have homogenized the description as the reviewer suggested. Site descriptions have been modified such that the same descriptors are indicated for all sites (number of stations, location, climate type, vegetation, precipitation and soil moisture sensors) as follows:«

- *Tunisian site: The Merguellil site is located in central Tunisia (9°54 E; 35°35 N). This site is characterized by a semiarid climate with highly variable rainfall patterns (average equal to 300mm/year), very dry summer seasons, and wet winters. The Merguellil site represents an agricultural region where croplands, namely, olive groves and cereal fields, prevail (Zribi et al., 2021). At this study site, a network of continuous thetaprobe stations installed at bare soil locations provided moisture measurements at depths of 5 and 40 cm. All measurements were calibrated against*

*gravimetric estimations. Data were obtained from the Système d'Information Environmental (SIE) web application catalog.*

- *Italian site: The Landriano site is located in northern Italy (Pavia province, Lombardia region). This station is located in a maize field, which was monitored in 2006 and from 2010 to 2011 (Masseroni et al., 2014). The average rainfall in Pavia province is of 650–700 mm and the climate is classified as 'Cfa' (cf. appendix A) and the field is irrigated by the border method with an average irrigation amount of approximately 100 to 200 mm per application with one to two applications per season due to the presence of a shallow groundwater table. Soil moisture measurements were performed with time domain reflectometer (TDR) soil moisture sensors. Five TDR soil moisture sensors were installed along a profile at depths of 5, 20, 35, 50, and 70 cm.*
- *Indian site: The Berambadi watershed is located in Gundalpet taluk, Chamarajanagara district, in the southern part of Karnataka state in India and covers an area of approximately 84 km²., The average rainfall is equal to 800 mm/year and the climate is classified as 'Aw' (cf. appendix A). Hydrological variables have been intensively monitored since 2009 in the Berambadi watershed by the Environmental Research Observatory ORE BVET and AMBHAS Observatory. The soil moisture levels at the surface (5 cm) and root zone (50 cm) are monitored with a HydraProbe sensor at different agricultural sites across the watershed, and in the current study, 4 stations were chosen.»*

Besides, the following climate and soil texture distributions for training, validation and test stations have been added in appendix B.

[Figure]

[Figure]

Reference to the appendix (cf. appendix A) is added whenever climate acronyms are used in the text.

**Comment:** "L162-164: Could you explain in more details why the pixel with highest NDVI value is selected among 16 days? Could you specify if it is at a single location, or if the pool of pixels to choose from covers a wider area than a single pixel?"

**Reply:** Actually, this is based on the MODIS Vegetation Indices product (MOD13Q1 version 6). This is mainly because NDVI is positive in cloud-free and dense vegetation areas and varies between 0.3 and 0.8, and can be negative over clouded scenes.

To further detail the algorithm of this product we could cite the Algorithm Theoretical Basis Document ATBD (The ATBD reference has been added in the revised manuscript): *«The construction of seasonal, temporal profiles requires a separate 'compositing' algorithm in which several VI images, over a given time interval (7-days, 10-days, etc.) are merged to create a single cloud-free image VI map with minimal atmospheric and sun-surface-sensor angular effects(…) The current procedure for generation of composited, AVHRR-based, NDVI products is the maximum value compositing (MVC) technique. This is accomplished by selecting, on a pixel by pixel basis, the input pixel with the highest NDVI value as output to the composited product. The procedure generally includes cloud screening and data quality checks.»*

**Comment:** "L165: Throughout the manuscript, the terms evapotranspiration and evaporation are used in alternance. It is often unclear if only soil evaporation or soil-plant evapotranspiration is concerned in the analysis. For instance, potential "evapotranspiration" is used to calculate "evaporation" efficiency, which is counter-intuitive. Could you clarify that and adjust the text accordingly? Do you separate "E" from "T"?"

**Reply:** We used a modified formulation of evaporation efficiency using potential evapotranspiration (PET) rather than potential evaporation. This approximation was applied in order to have a continuous estimation of evaporation efficiency during vegetated and non-vegetated contexts without resorting to the complex exercise of

separating evaporation and transpiration processes. In fact, PET remote sensing-based values are more readily available.

**Comment: "**L166-168: The expression "evaporation efficiency" has been mentioned a few times already, but it is unclear what it is as it has not been defined yet, which contributes to the overall confusion. Maybe a definition in a few words would be good early on, for instance in the introduction, which is supposed to introduce important concepts for the reader to understand them before going through the bulk of the manuscript. This remark could apply to other important concepts in the manuscript."

**Reply:** We agree with the reviewer on this point. The evaporation efficiency should have been defined earlier in the text. In the revised paper, the definition has been added in  section (2.1.5 Potential evapotranspiration): *«Similarly, we assessed the impact of considering a remote sensing-based evaporation efficiency, which is initially defined as the ratio of actual to potential soil evaporation,on RZSM prediction.»*

**Comment: "**L174: If the raw PET value is the "sum" of PET over 8 days, do you turn them into daily values by dividing them by 8? Please clarify the integration and deconvolution process."

**Reply:** Indeed. We linearly interpolated the 8-day PET product so that we obtain PET daily values that we divided after by 8. In the revised paper, we have clarified this as follows: *«To obtain daily PET values, we performed a linear interpolation over the 8-day product and then we divided by eight the interpolated value.»*

**Comment: "**L182: The SWI_m index is calculated as a recursive series, but it is not mentioned how the first value of SWIm is calculated in the time series. Please explain it."

**Reply:** We thank the reviewer for bringing this to our attention. This important point has been added in the revised paper. As described in (Albergel et al., 2008) :*«For the initialisation of the filter, gain $K_1$ =1 and $SWI_{t1}$ =ms($t_1$) (first value of scaled SSM).»*

**Comment: "**L182-189: It is unclear what the subscript "m" stands for in SWI_m, and is it necessary?"

**Reply:** We agree with the reviewer. Subscript m is not necessary and has been deleted.

**Comment: "**Overall many of the symbols feel a bit obscure or poorly chosen. A few examples include the symbol \beta_3 (Why is there a subscript 3, and is it necessary? Is it the same variable as B_3? If not, please choose symbols that are easier to distinguish , A_3 (Why is there a subscript 3, and is it necessary?, the variables that have a temporal component (The time indices alternate between "t_n" and "n", please be more consistent), \theta_L (Please do not use soil layer thickness

as a layer index as two layers with the same thickness would not be distinguishable).”

**Reply:** Actually, $\beta 3$ is different from B3. $\beta 3$ refers to evaporation efficiency while B3 is a tuning parameter. The subscript 3 is used in reference (Merlin et al.,2010) who tested 3 models and used parameters A and B for each. Subscripts (1,2,3) refer to the considered model. Here, we used the equations of model 3. In order to clarify for the reader, in the revised paper, we mentioned that we are using equations of model 3 by (Merlin et al.,2010) and deleted subscript 3.

In the revised paper, section 2.2.2 (Evaporation efficiency) now reads:

*«An ANN model with evaporation efficiency input was also developed. This variable, which is defined as the ratio of the actual to potential soil evaporation, was first introduced in (Noilhan, J. and Planton, 1989; Jacquemin et al., 1990; Lee et al., 1992) and thereafter readapted in (Merlin et al., 2010) to include the soil thickness. In our work, we use a modified evaporation efficiency formulation, based on the third model developed in (Merlin et al., 2010), which can be expressed as follows (cf. appendix C):*

$$\beta = [\tfrac{1}{2} - \tfrac{1}{2}\cos(\pi\theta/\theta_{max})]^{P*}$$

*(3)*

*»*

**Comment:** "L182-189 and throughout the manuscript: Please introduce new variables with their units.”

**Reply:** We thank the reviewer for pointing this out. Missing units have been added in the revised paper.

**Comment:** "L195-200: This paragraph should be located in the Results or Discussion section.”

**Reply:** We agree with the reviewer. In the revised paper, this paragraph has been moved to the results section. Section *«3.1 Exponential filter characteristic time length»* has been added.

**Comment:** "L201-208: These paragraphs should be located in the Discussion section.”

**Reply:** In the revised paper, we will move this paragraph to the results section, as we want to concentrate the discussion section on the outputs related to the paper objectives in a broader perspective.

**Comment:** "L214-229: I find the presentation of the evaporative efficiency equations confusing. The text suggests that \beta_3 is the evaporation efficiency index, but it also says that it is the ratio of actual to potential soil evaporation , which does not seem to be the content of the equations. Please clarify directly that equation 4 is not

used or even better, put it in appendix (maybe describe the connection between the standard equation of P and yours in appendix, in order to go to the point in this part of the Methods). If P* is used in the calculation of \beta_3, then please use the symbol P* in equation 3. Also, there is no need to provide two definitions of \theta_max then to say that you are not using the first one. Only mention in the Discussion section that another definition exists if you think it is worth discussing, or else send this information to appendices. Same story for the choice between LE_p and PET."

**Reply:** Evaporation efficiency is analytically defined as the ratio of actual to potential soil evaporation. New analytical models were later suggested in literature such as in (Merlin et al.,2010) who proposed new formulations which fill the gap of the first models.

We agree with the reviewer that the equations need to be more clarified. In the revised paper, we have clarified the variables and symbols we used to compute evaporation efficiency. Standard equations as they appear in (Merlin et al., 2010) have been added to appendix C. In the revised paper, section 2.2.2 now reads:

«

$$\beta = [\tfrac{1}{2} - \tfrac{1}{2}\cos(\pi\theta/\theta_{max})]^{P*}$$

(3)

*where: -* $\beta$ *is evaporation efficiency*

   *-* $\theta$ *is the water content in the considered soil layer of thickness L.*

   *-* $\theta$*max is the maximum soil moisture at each station.*

   *- P\* is a parameter computed as follows:*

$$P^* = \frac{PET}{2B}$$

(4)

   *P\*, a proxy of parameter P (cf. appendix C), represents an equilibrium state controlled by retention forces in the soil, which increase with the thickness L of considered soil and by evaporative demands at the soil surface.*

   *-PET is the potential evapotranspiration (PET) extracted from the MODIS 500-m 8-day product (MOD16A2).*

*The soil evaporation efficiency computed by model 3, developed in (Merlin et al., 2010), decreases when PET increases. Retention force and evaporative demand make the term P increase (replaced by P\*), as if an increase of potential evaporation LE_p (here replaced by PET) at the soil surface would make the retention force in the soil greater.*

*Merlin et al. (2010) tested this approach at two sites in southwestern France using in situ measurements of actual evaporation, potential evaporation, and soil moisture at five different depths collected in summer.*

*Model 3 was able to represent the soil evaporation process with a similar accuracy as the classical resistance-based approach for various soil thicknesses up to 100 cm. Merlin et al. (2010) affirm the parameterization of P as function of $LE_p$ (here PET) indicates that $\beta$ cannot be considered as a function of soil moisture alone since it also depends on potential evaporation. Moreover, the effect of potential evaporation on $\beta$ appears to be equivalent to that of soil thickness on $\beta$. This equivalence is physically interpreted as an increase of retention forces in the soil in reaction to an increase in potential evaporation.»*

**Comment:** "L250: Please clarify in this part of the Methods how you are using the additional independent datasets from Italy, India and France."

**Reply:** These datasets (Italy, India and Tunisia) are used as new independant test locations external to the ISMN database that was in the first part used for training, validation and test. In the revised paper, the text reads: *«In a second step, tests were conducted on data external to the ISMN database namely on sites of Tunisia, Italy and India. The trained models over ISMN are used only in prediction mode over these sites. The data for SSM in addition to the other features are used as inputs and RZSM is predicted in outputs.»*

**Comment:** "Results

Figure 3: The caption does not specify if the results displayed concern the training or validation dataset (or both) . The evaluation of the quality of the prediction should be done on the validation dataset (please display these results independently in parallel to the results of the training dataset to ease the comparison),whose quality of fit should remain similar to the quality of fit of the training dataset. If the quality of fit in the validation stage is substantially lower than in the training stage, the ANN is considered to "overfit" the training dataset. I do not think that this comparison is provided in the current version of the manuscript, while it is central for the evaluation of the results. There is an evaluation of the quality of fit in an independent dataset (for Italy, France and India) but the quality of fit of this "second validation" step is not compared to the quality of fit during the training stage. Also, the correlation between measurements and predictions as an indicator of quality of fit has limitations. One of them is that it is "blind" in case of predictions proportional to measurements. That is why root mean square errors are provided systematically when conducting such a model validation. Please systematically provide results for this metrics too."

**Reply:** We understand that the following comment contains three interlinked parts. We provide here the answers for these three parts:

1/ Separation of training, validation and test datasets for the ISMN exercice :

In the revised paper, new histograms have been added to separate between training, validation and test datasets. Similarly, rates of correlation improvement have been

specified for each type of dataset separately. In the revised paper, section 3.2 (Intercomparison of the ANN models) now includes new figures and reads:

[revised manuscript text omitted]

2/ Comparison between the independent dataset quality of predictions and training quality of fit over ISMN.

We added a table with performance metrics (correlation and RMSE) for training stations with a similar climate type than stations used in the external validation step (Tunisia). Results with both models ANN_SSM and ANN_SSM_NDVI_EVAP-EFF-B60_EXP-FILT-T5 are provided. In the revised paper, section (3.3 Robustness of the approach) now reads:

*«However, the consideration of additional features, namely, the NDVI, evaporation efficiency and SWI in the ANN models resulted in a good agreement between the in situ and predicted RZSM values (Fig. 4). The correlation values were improved by 60.04%, 169.5%, 112.02%, 80.23% and 53.7% at stations Barrouta-160, Hmidate_163, Barrage_162, Bouhajla_164 and P12, respectively, with the ANN_SSM_NDVI_EVAP-EFF-B60_EXP-FILT-T5 model over ANN_SSM model values. Similarly, RMSE values were reduced (Table 5). As shown in figure 4, the most complex ANN model is able to capture the variations of RZSM. This finding highlights the added value of our hybrid approach based on an association of a machine learning method with process-related variables. Instead of injecting uncertain information in physical models, such as soil properties, we used a nonparametric method related to physical processes without using forcing data that may be subject to errors and potentially lead to inaccurate tracking of the long-term evolution of soil moisture.*

[Figure]

**Figure 4.** *In situ SSM, in situ RZSM, and predicted RZSM series at the stations in the Kairouan Plain (Tunisia) with model ANN_SSM_NDVI_EVAP-EFF-B60_EXP-FILT-T5 (cf. appendix G for larger figure format).*

*A second comparison can be conducted between the quality of fit of these independent datasets and training datasets. Actually, the climate class of the Tunisian stations is 'Bsh' (Arid Steppe Hot, cf. appendix A). At the training stage, no station falls into the climate class 'Bsh' (Arid Steppe Cold, cf. appendix A). However, some training stations fall under a similar climate class which is 'Bsk' (cf. appendix B). Table 5 presents correlation and RMSE values for these training stations and Tunisian sites with both models ANN_SSM and ANN_SSM_NDVI_EVAP-EFF-B60_EXP-FILT-T5. For all training stations, performance metrics are slightly enhanced with the most complex ANN model compared to reference model ANN_SSM, except for stations GrouseCreek, Harmsway and Lind#1 which performance decreases. Overall, the range of correlation values is similar for training and external validation stations with model ANN_SSM_NDVI_EVAP-EFF-B60_EXP-FILT-T5 and RMSE is well reduced for Tunisian stations compared to training stations. Given the results on unseen datasets, namely on Tunisia, the performance of the most complex ANN model is good as it is able to generalize the patterns present in the training dataset.*

**Table 5.** *Performance metrics of models ANN_SSM and ANN_SSM_NDVI_EVAP-EFF-B60_EXP-FILT-T5 at training stations of climate "Bsk" and Tunisian stations of climate "Bsh".*

| *Model* | *ANN_SSM* | | *ANN_SSM_NDVI_EVAP-EFF-B60_EXP-FILT-T5* | |
|---|---|---|---|---|
| | *Training stations (climate class 'Bsh')* | | | |
| *Station* | *Correlation* | *RMSE* | *Correlation* | *RMSE* |

| | | | | |
|---|---|---|---|---|
| Banandra (OZNET) | 0.701 | 0.05 | 0.764 | 0.046 |
| DRY-LAKE (OZNET) | 0.674 | 0.031 | 0.692 | 0.03 |
| CPER (SCAN) | 0.691 | 0.032 | 0.695 | 0.032 |
| EPHRAIM (SCAN) | 0.758 | 0.051 | 0.791 | 0.046 |
| GrouseGreek (SCAN) | 0.818 | 0.033 | 0.802 | 0.035 |
| HarmsWay (SCAN) | 0.705 | 0.034 | 0.622 | 0.038 |
| Lind#1 (SCAN) | 0.605 | 0.055 | 0.483 | 0.022 |
| *External test stations (Tunisia)* | | | | |
| Station | Correlation | RMSE | Correlation | RMSE |
| Barrouta_160 | 0.463 | 0.021 | 0.714 | 0.016 |
| Hmidate_163 | 0.318 | 0.019 | 0.834 | 0.011 |
| Barrage_162 | 0.416 | 0.035 | 0.864 | 0.019 |
| Bouhajla_164 | 0.435 | 0.016 | 0.733 | 0.01 |
| P12 | 0.581 | 0.047 | 0.861 | 0.029 |

*»*

3/ Choice of fit metrics :

It is true that one indicator of fitness is limited.  We chose not to show RMSE as it involves 2 components, namely bias and correlation. In order to give a good estimate of bias, we need to scale the soil moisture based on soil texture which is not very precise. Also, we need to estimate the pedo-transfer functions of the soil based on soil texture which are not very precise as aforementioned. Nevertheless, we have added supplementary materials of RMSE in appendix D as follows:

[Figure]

[Figure]

*RMSE histograms of all tested ANN models compared to ANN_SSM (a) on training stations (b) on validation stations (c) on test stations*

**Comment: "**L255-258 and other places in the results: If the validation and training sets seem are merged in the histograms, it is possible that the improvement of the quality of fit is due to stronger "overfitting" allowed by the larger number of ANN parameters as more input variables types are added from the simplest ANN_SSM to the most complex ANN_SSM_NDVI_EVAP-EFF-B60_EXP-FILT_T5 model. If that is the case, from what is shown the authors cannot exclude that the improved quality of fit is due to overfitting allowed by the increasing degree of freedom. Could you specify the degree of freedom of each ANN model, and show results for training and validation stages separately?"

**Reply:** In the revised paper, the histograms have been separated as answered in the previous comment.

The number of parameters of each network was added to Table 1. The highest number of parameters corresponds to the most complex model i.e ANN_SSM_NDVI_EVAP-EFF-B60_EXP-FILT-T5 and is equal to 161 parameters versus 101 parameters for the least complex model ANN_SSM.

*Table 1. ANN model configurations with the respective input variables;\*: rolling averages of SSM over 10 days; \*\*: rolling averages of SSM over 30 days; \*\*\*: rolling averages of SSM over 90 days; \*\*\*\*: number of parameters of the ANN model.*

| Model / Features | SSM_10d_RAV[*] | SSM_30d_RAV[**] | SSM_90d_RAV[***] | SST | NDVI | SWI | EVAP | Nb[**][**] |
|---|---|---|---|---|---|---|---|---|
| ANN_SSM | X | X | X | | | | | 101 |
| ANN_SSM_TEMP | X | X | X | X | | | | 121 |
| ANN_SSM_NDVI | X | X | X | | X | | | 121 |
| ANN_SSM_EXP-FILT_T5 | X | X | X | | | X | | 121 |
| ANN_SSM_EVAP-EFF_B60 | X | X | X | | | | X | 121 |
| ANN_SSM_NDVI_EVAP-EFF-B60_EXP-FILT-T5 | X | X | X | | X | X | X | 161 |

**Comment:** "L288-311: It is interesting that for each ANN, the set of parameters remains the same across stations while also allowing seemingly good predictions of RZSM. I think it is something to discuss, particularly because it does not have to be so. The authors could have separated the datasets in sub-groups corresponding to a few major climate types or major soil types. Instead, they merged them all, seemingly trying to derive "general relations" between RZSM and a few variables, regardless of the soil and climate types. I think it would be of interest to develop the analysis (or at least the perspectives) in this direction. Though it is not within the scope of the objectives of the manuscript."

**Reply:** In a previous study we conducted (Souissi et al.,2020) and that was cited in this paper, we evaluated the accuracy and transferability of artificial neural networks in predicting in Situ root-zone soil moisture for various regions across the globe using only in situ surface soil moisture (SSM). We also addressed the transferability of the ANN model across climatic and soil texture conditions.

Figure 6 also partially tackles this question by showing the most representative inputs across climate classes. As suggested by the reviewer, this point is of interest and can be studied in future work. We added it as a perspective in the conclusion section that now reads: *«As a research perspective, datasets can be separated in clusters corresponding to major climate classes and/or soil types. More analysis can be conducted in this direction to eventually make connections between the different inputs and climate/soil configurations.»*

**Comment:** "Figure 4: This is the only time SSM data (measured) and RZSM data (measured and predicted with only one of the ANNs) is shown. This is far too limited to have an idea of the quality of predictions. Please provide at least a comparison of

ANN_SSM and ANN_SSM_NDVI_EVAP-EFF-B60_EXP-FILT_T5 models within the Results section, and more diverse examples (e.g., training vs validation datasets, a few typical fits, good and bad) in the Results and/or appendices."

**Reply:** Time series of good and less good quality of fit have been added in appendix E for training, validation and test stations separately using models ANN_SSM and ANN_SSM_NDVI_EVAP-EFF-B60_EXP-FILT-T5 . Appendix E now reads:

*«Training stations:*

*Station 'Beloufoungou Mid' (network 'AMMA-CATCH')*

| ANN_SSM | ANN_SSM_NDVI_EVAP-EFF-B60_EXP-FILT-T5 |
|---|---|
|
[Figure]
 | |

*Station 'HarmsWay' (network 'SCAN')*

| ANN_SSM | ANN_SSM_NDVI_EVAP-EFF-B60_EXP-FILT-T5 |
|---|---|
|
[Figure]
 | |

*Validation stations*

*Station 'Cabrieres D'Avignon' (network 'SMOSMANIA')*

| *ANN_SSM* | *ANN_SSM_NDVI_EVAP-EFF-B60_EXP-FILT-T5* |
|---|---|

[Figure]

*Station 'Nephi' (network 'SCAN')*

| *ANN_SSM* | *ANN_SSM_NDVI_EVAP-EFF-B60_EXP-FILT-T5* |
|---|---|
|
[Figure]
 | |

*ISMN test stations*

*Station 'Wankama' (network 'AMMA-CATCH')*

| *ANN_SSM* | *ANN_SSM_NDVI_EVAP-EFF-B60_EXP-FILT-T5* |
|---|---|
|
[Figure]
 | |

*Station '2.04' (network 'HOBE')*

| ANN_SSM | ANN_SSM_NDVI_EXP-T5_EVAP-B60 |
|---|---|

[Figure]

**Comment:** "Figure 5: It is unclear to me why the bars corresponding to "all stations" do not have the same length across panels (except for panel d, ANN_SSM_TEMP, which does not have temperature data for all stations). I think this should be clarified. Also, it is unclear what the numbers in blue are on top of the bars. This should appear in the caption and body of the text".

**Reply:** Actually, as the temperature values used are in-situ values we have practically this information for all stations except for some. For other variables (remote-sensing based), we do not have data for all stations because of the spatial sampling. Here, we chose to show the best-performing models with the most available data. We could have selected only common stations where all input data are available. However, our objective is to prepare the work for RZSM spatial mapping and to be the most accurate as possible.

As for the numbers in blue, they are marked with a symbol '*' that is defined at the bottom of the plot as '*mean correlation change rate per climate class" and are computed using equation 5.

**Comment:** "L337-364: These are new results. They should appear in the Results section even if they are a transverse analysis of results already shown. The associated methods (including equation 6) should be detailed in the Material and Methods section. There is substantial space to develop a discussion in view of the existing literature, and a few perspectives."

**Reply:** Our objective here was to discuss the results of the performance of each ANN model with respect to climate regions either by showing the improvement in the form of histograms or a global map. For this reason, we decided to put them in the discussion section.

**Comment:** "Figure 6: The climate types in the legend should be defined through a reference to the table in appendix. The darkest colours do not allow reading the overlapping symbols for the types of ANN. Please display results in a way that makes it comfortable to read them, possibly by duplicating panels. It would also be interesting to see where the stations are located, using colours to show which input variable was the most critical for each site, and compare with the "climate-type-associated-critical-input" already there in the figure. Please make the caption more descriptive (in general across figures). A first point is to mention in the caption that what is shown is a "World map of (…)"."

**Reply:** To make it easier to read, we have regenerated the map so that hatches are more visible. In the revised paper, reference to appendix A has been added in the caption of figure 6. Figure 6 with revised caption is as follows:«

[Figure]

***Figure 6.*** *World map of best-performing ANN models per climate class based on the mean correlation change rate; colors correspond to climate classes (cf. appendix A), hatches correspond to the most contributive input to the predictions namely: EVAP (evaporation efficiency), EXP (exponential filter SWI), NDVI , TEMP (surface soil temperature).»*

**Comment:** "L344-364: This part of the analysis uses substantial extrapolation spatially-speaking, which I think would be ok if it was shown that within each climate type, one of the input variable type clearly stands out as a key predictor. Given the results shown in Figure 5, it seems that it is not the case, and that various stations allocated to the same climate type may most rely on diverse input variables. Thus, I am not really convinced about the added value of this part of the analysis (at the end of the discussion, the authors also acknowledge that this classification suffers limitations), and I wonder if the authors could make it more convincing using

complementary analyses, or else I think it would be better to remove it from the paper."

**Reply:** As said by the reviewer, we have already mentioned that this classification suffers limitations. However, the results are quite promising and may suggest further investigations in future works.

**Comment:** "L377-383: Please acknowledge in the text that the quality of fit decreased in Italy and India. This is also worth developing in the Discussion section."

**Reply:** The change in correlation over India and Italy is about -0.04 in correlation which is nonsignificant. We clarified this point as follows in the revised paper : *«In India and Italy, the correlations were already high with the reference model ANN_SSM. The change in correlation after the addition of process-related features, namely NDVI, is about -0.04 which is nonsignificant, and is potentially because of the cloudy conditions in India and noisy MODIS products. Also the crop heterogeneity and sample impurity makes MODIS NDVI products not adapted to all sites.»*

Also, limitations of MODIS NDVI were added in section 3.3 Robustness of the approach as follows:

*«The presence of clouds in the MODIS NDVI and potential evapotranspiration products could explain this observation at sites of South-India and North-Italy. In South-India, for instance, the maximum variability in soil moisture occurred during the monsoon season, which is characterized by a large amount of clouds. Moreover, the coarse resolution of MODIS NDVI product makes it sometimes not adapted to the considered site. (Chen et al., 2016) investigated the impact of sample impurity and landscape heterogeneity on crop classification using coarse spatial resolution MODIS imagery. They showed that the sample impurity such as mixed crop types in a specific sample, compositional landscape heterogeneity that is the richness and evenness of land cover types in a landscape, and configurational heterogeneity that is the complexity of spatial structure of land cover types in a specific landscape are sources of uncertainty affecting crop area mapping when using coarse spatial resolution imagery. High resolution NDVI from sensors like Sentinel-2 could have been used in this exercise to mitigate the spatial resolution issue, however, MODIS data were privileged in order to provide NDVI and PET from the same sensor.»*

**Typos and minor details:**

**Comment:** "L35, 81, 88 & 334: The acronyms ECV and ML are introduced but I couldn't find them in the rest of the text. The acronym MLP hasn't been defined yet when first introduced and is only used once after being defined. I could not find the definition of the acronym LST. More generally I would avoid introducing unnecessary acronyms, and remove the ones listed in this comment."

**Reply:** ML acronym was deleted. We kept ECV because it is a widely used term in the community.

We replaced MLP by ANN in introduction, and removed the acronym from materials and methods.

LST was removed.

**Comment: "**L41: I don't get why the terms mission is there twice, while it seems the sentence describes sensors. Please correct or clarify."

**Reply:** We agree with the reviewer. The word "mission" has been removed.

**Comment: "**L43: has -> have"

**Reply:** We have corrected it.

**Comment: '**L68, 69, 74 and many other instances: When the authors of the cited paper are the subject of the sentence, or any kind of complement within the sentence, please use the following form "author et al. (year)" rather than "(author et al., year)"."

**Reply:** We agree with the reviewer. In the revised paper, we have replaced "(author et al., year)" by "author et al. (year)" when necessary.

**Comment: "**L77: Please clarify that these are correlations between series of measured and predicted RZSM and whether it concerns the training stage or validation on independent data."

**Reply:** We agree with the reviewer. It has been specified in the revised paper.

**Comment: "**L87-90: In point (1), the temperature input is missing in the list."

**Reply:** We agree. Surface soil temperature has been added as follows: *"we extend the feature list to include NDVI time series, surface soil temperature and process-related variables, namely, the soil water index given by a recursive exponential filter and remote sensing-based evaporation efficiency".*

**Comment: "**L176: The sub-title "Methods" in the section Material and Methods is not very specific. Could you modify it to make it more specific?"

**Reply:** The use of sub-title "methods" is very standard in scientific articles.

**Comment: "**L186: "which occurs in [0, 1]" -> "which ranges between 0 and 1". "

**Reply:** It has been corrected.

**Comment: "**L191-192: The expression "given (…) T values" appears twice in the same sentence."

**Reply:** We agree. «Given each T value» has been removed

**Comment: "**L210: It seems to me that "two" ANN models include the evaporative efficiency input."

**Reply:** At that part of the manuscript, we were introducing models with only one process-related input (in addition to SSM). The most complex model with three process-related inputs, amongst which is evaporation efficiency, was described later.

**Comment: "**L255-257: No need for 2 digits in the numbers. One or none would seem sufficient to me given the large differences."

**Reply:** We kept two digits to be consistent all over the text.

**Comment: "**Throughout figures: Please use letters to refer to the panels within individual figures."

**Reply:** We agree. Letters have been added for each panel in figures 3 and 5.

**Comment: "**L342: There is a "*" symbol at the start of equation 6, which seems to be a typo."

**Reply:** It is not a typo. Actually, the "*" is used in Figure 5 next to each mean correlation change rates (in blue) which were computed using equation 5.

---

## Author Comment (AC2)

**Response to comments of Referee #2**

Please find in Black the reviewer's comments and in blue our answers.

**Comment:** "This paper discusses the relationship between surface moisture and moisture at depth in the soil layer occupied by roots. This is an important issue for analyzing the time series of surface moisture observed by satellite by providing much more relevant information to characterize the functioning of ecosystems. Indeed, deep moisture has a very important impact on water fluxes by controlling both transpiration and deep drainage. The proposed approach is to use neural networks trained on a large dataset as offered by the ISMN. The main innovation is to introduce variables into the network training that can account for factors impacting the relationship between surface moisture and humidity in the RZSM. I find that the introduction does not insist enough on the processes that govern this relationship. Indeed, it is strongly linked to 1) root uptake, which depends on the canopy and the root profile in interaction with the climate and 2) capillary rise, which means that, depending on the properties of the soil, the water that evaporates from the surface layer is more or less compensated. This very important process, especially for lightly covered soils, is never mentioned. This knowledge of physical processes could have been put forward to justify the process based variables."

**Reply:** We would like to thank the reviewer for his constructive comments that helped us enhance the quality of the paper.

We agree with the reviewer that the aforementioned processes govern the relationship between SSM and RZSM. In the revised paper, the introduction has been updated to describe in summary the relation governing SSM and RZSM as follows :*«RZSM is nonlinearly related to SSM through different hydrological processes, such as diffusion processes. The root-zone soil moisture may be extracted by evaporation at the surface, through root extraction or by capillary rises (Calvet et Noilhan, 2000).»*

**Comment:** "I remain a little dubious about the choice of process based variables."

**Reply:** We changed the term «process-based» to «process-related» in order to avoid any confusion.

**Comment:** "NDVI: for me there is no doubt that this variable must be taken into account. On the other hand, the use of ndvi modis variable does not seem to me to be adapted to the sites used. Indeed, many measuring stations are placed on sites where the vegetation is not representative of the nearby environment. SMOSMANIA is placed on a meteorological station with a non-irrigated fallow land placed in the middle of an agricultural zone (probably dominant at the scale of the modis pixel). The stations of the plain of Kairouan are on bare soil (probably to simplify the management of sensors) while the plain is an agricultural area. So I think there may be a big difference between the modis ndvi and the ndvi on the representative area

of the measurement. This is illustrated in table 4 where the model including the ndvi led to degraded results in comparison to ANN-SSM."

**Reply:** We agree with the reviewer that it is important to consider NDVI and also that the use of MODIS NDVI has a scale mismatch with point observations. Actually, higher resolution optical remote sensing products at high revisit are available (i.e. Sentinel-2 NDVI), but we privileged the MODIS dataset in order to combine the NDVI and the potential evapotranspiration from the same plateforme. In the revised paper, we have added a paragraph to discuss this point as follows:

*«The presence of clouds in the MODIS NDVI and potential evapotranspiration products could explain this observation at sites of South-India and North-Italy. In South-India, for instance, the maximum variability in soil moisture occurred during the monsoon season, which is characterized by a large amount of clouds. Moreover, the coarse resolution of MODIS NDVI product makes it sometimes not adapted to the considered site. (Chen et al., 2016) investigated the impact of sample impurity and landscape heterogeneity on crop classification using coarse spatial resolution MODIS imagery. They showed that the sample impurity such as mixed crop types in a specific sample, compositional landscape heterogeneity that is the richness and evenness of land cover types in a landscape, and configurational heterogeneity that is the complexity of spatial structure of land cover types in a specific landscape are sources of uncertainty affecting crop area mapping when using coarse spatial resolution imagery. High resolution NDVI from sensors like Sentinel-2 could have been used in this exercise to mitigate the spatial resolution issue, however, MODIS data were privileged in order to provide NDVI and PET from the same sensor.»*

A sentence has also been added to the conclusion as follows: *«In India and Italy, the correlations were already high with the reference model ANN_SSM. The change in correlation after the addition of process-related features, namely NDVI, is about -0.04 which is nonsignificant, and is potentially because of the cloudy conditions in India and noisy MODIS products. Also the crop heterogeneity and sample impurity makes MODIS NDVI products not adapted to all sites.»*

**Comment:** "The evaporative fraction as calculated is directly related to the surface moisture. There is therefore no introduction of information except via LEP which acts as a  second order factor on the evaporative fraction."

**Reply:** We agree. Even though the evaporation efficiency formulation is related to surface soil moisture, it is still a new source of information for the neural network via the PET normalization. As mentioned by the reviewer, it acts as a second order factor in the analytical model.

The text related to this section was updated as follows based on the all the reviewers comments:

 *«-P\*, a proxy of parameter P (cf. appendix C), represents an equilibrium state controlled by retention forces in the soil, which increase with the thickness L of considered soil and by evaporative demands at the soil surface.*

*-PET is the potential evapotranspiration (PET) extracted from the MODIS 500-m 8-day product (MOD16A2).*

*The soil evaporative efficiency predicted by model 3 by (Merlin et al., 2010) decreases when PET increases. Retention force and evaporative demand make the term P increase (replaced by P\*), as if an increase of LEp (here PET) at the soil surface would make the retention force in the soil greater. Merlin et al. (2010) tested this approach at two sites in southwestern France using in situ measurements of actual evaporation, potential evaporation, and soil moisture at five different depths collected in summer. Model 3 was able to represent the soil evaporation process with a similar accuracy as the classical resistance-based approach for various soil thicknesses up to 100 cm. Merlin et al. (2010) affirm the parameterization of P as a function of LEp (here PET) indicates that $\beta$ cannot be considered as a function of soil moisture alone since it also depends on potential evaporation. Moreover, the effect of potential evaporation on $\beta$ appears to be equivalent to that of soil thickness on $\beta$. This equivalence is physically interpreted as an increase of retention forces in the soil in reaction to an increase in potential evaporation.»*

**Comment: "**The recursive exponential filter completes the filtering by averaging over 10 30 and 90 days. It would have been interesting to compare them in order to identify to what extent these filters are complementary."

**Reply:** In order to clarify, we didn't apply the recursive exponential filter on the 10, 30 and 90-day averages.We have separated inputs for the 10, 30 , 90 day rolling averages and the recursive exponential filter outputs. This being said, in (Souissi et al, 2020) we investigated the impact of temporal parametrization of SSM inputs, namely the use of hourly, daily or rolling averages over 10, 30 and 90 days. In this paper, we were focused on the study of the impact of the exponential filter which has been identified as a simplified analytical solution for RZSM prediction.

**Comment: "**The surface temperature could have been an indicator of the evaporative intensity. However, below the vegetation cover, the interpretation is far from obvious and requires knowledge of the air temperature in the canopy. Here it is not clear at what depth it is measured (probably at the depth of the first sensor). In this case the only interest seems to me to be to be able to flag the periods of freezing to eliminate the data which do not have a physical meaning. I would make this cleaning before training the neural networks."

**Reply:** We thank the reviewer for this point. We have made the following clarification in the ISMN soil moisture dataset description: *«For each selected station, the root zone soil moisture observation point is located between 30 and 55cm (Table 2). For each soil moisture hourly acquisition, ISMN provides quality flags. Quality flags can be marked as 'C' (exceeding plausible geophysical range),' D' (questionable/dubious), 'M' (missing), or 'G' (good) (Dorigo et al.,2011). Category 'D' has subset flags namely 'D01' for which in situ soil temperature < 0°C, 'D02' that flags points at which in situ air temperature < 0°C as well as 'D03' that also flags areas where GLDAS soil temperature < 0°C. In our study, only soil moisture data with a 'G' labeled quality flag 'G' were retained.»*

**Comment:** "A more thorough discussion of the process based variables would be necessary, showing in particular on examples how they allow a better understanding of the relationship between surface and depth."

**Reply:** We agree with the reviewer. More results and discussions have been added to the manuscript. Time series have also been added in appendix E to compare the quality of fits of the least and the most complex models. As also inquired by reviewer 1, we have enriched the results section. In the revised paper, section (3.2 Intercomparison of the ANN models) has been modified such that there is a separation of training, validation and test datasets. Also, a comparison between the independent dataset quality of predictions (Tunisia) and training quality of fit over ISMN has been conducted. Section 3.2 now reads:

[revised manuscript text omitted]

**Comment: "**I would now focus on the results. Are the results presented in table 3 qualitatively good? For example, for the RMSE, the introduction of co-variables has a positive impact in only 57% of cases at best. This also means that in 43% of the cases the results are worse. I think that a more rigorous statistical analysis would be necessary to decide whether or not the gain is significant."

**Reply:** In the revised paper, the text has been modified to separate between training, validation and test stations as suggested by reviewer 1. New rates were provided accordingly. Indeed, not all stations undergo an improvement when process-related

variables are added. As we have clarified in the revised text, a small percentage of stations undergo a decrease in correlation of more than 0.1 and an increase in RMSE of more than 0.01. Section 3.2 Intercomparison of ANN models now reads:

*«Considering model ANN_SSM_NDVI_EVAP-EFF-B60_EXP-FILT-T5, only one training station had a decrease in correlation by more than 0.1 namely station 'Lind#1' (network 'SCAN') compared to reference model ANN_SSM. All inputs were not available at the same dates which implied a significant reduction in data points (cf. appendix F). The decrease in correlation and increase in RMSE didn't exceed 0.1 and 0.01 m3/m3 for the rest of stations of lower performance metrics with the most complex ANN, respectively.*

*Similarly for validation stations, only one station had a decrease in correlation above 0.1, namely station 'PineNut' (network 'SCAN'), with model ANN_SSM_NDVI_EVAP-EFF-B60_EXP-FILT-T5. This decrease can be also explained because of data shortage (cf. appendix F). The decrease in correlation and increase in RMSE didn't exceed 0.1 and 0.01 m3/m3 for the rest of stations of lower performance metrics with the most complex ANN, respectively.*

*Regarding test stations , correlation decrease by more than 0.1 and RMSE increase by more than 0.01 m3/m3 with model ANN_SSM_NDVI_EVAP-EFF-B60_EXP-FILT-T5 compared to model ANN_SSM was detected for only 2 stations.  Both stations, namely station 'S-Coleambally' and 'Widgiewa' which belong to network 'OZNET', significantly lose in data volume when process-related variables are integrated in ANN and more precisely because of NDVI data availability (cf. appendix F). For the rest of test stations, correlation decreased and RMSE increased simultaneously by less than 0.1 and 0.01 m3/m3 with model ANN_SSM_NDVI_EVAP-EFF-B60_EXP-FILT-T5, respectively.*

*Table 3. Proportion of the stations which performance enhances using the ANN models enriched with process-related features compared to model ANN_SSM (\*: % of stations at which the correlation improves over the model ANN_SSM level; \*\*: % of stations at which RMSE improves over the model ANN_SSM level)*

| Model | Training stations | | Validation stations | | Test stations | |
|---|---|---|---|---|---|---|
| | % of stations (corr ↑)* | % of stations (RMSE ↓)** | % of stations (corr ↑)* | % of stations (RMSE ↓)** | % of stations (corr ↑)* | % of stations (RMSE ↓)** |
| ANN_SSM_NDVI | 65.82 | 44.3 | 45.71 | 40.0 | 55.22 | 40.3 |
| ANN_SSM_TEMP | 49.4 | 25.3 | 55.56 | 38.89 | 59.35 | 42.99 |
| ANN_SSM_EXP-FILT-T5 | 64.56 | 36.71 | 60.61 | 42.42 | 63.68 | 50.25 |
| ANN_SSM_EVAP-EFF-B60 | 54.55 | 28.57 | 52.94 | 41.18 | 52.33 | 48.19 |
| ANN_SSM_NDVI_EVAP-EFF-B60_EXP-FILT-T5 | 84.06 | 62.32 | 61.29 | 54.84 | 62.07 | 54.02 |

*Table 4. Proportion of the stations which correlation decreases using the ANN models enriched with process-related features compared to model ANN_SSM (\*$\Delta_{corr}$=$corr_{ANN\_SSM}$ − $corr_{ANN\_SSM\_X}$, X denotes a or a combination of process-related variables)*

| Model | Training stations | | Validation stations | | Test stations | |
|---|---|---|---|---|---|---|
| | % of stations corr ↓ and 0.05<$\Delta_{corr}$\*< 0.1 | % of stations corr ↓ and $\Delta_{corr}$\*>0.1 | % of stations corr ↓ and 0.05<$\Delta_{corr}$\*< 0.1 | % of stations corr ↓ and $\Delta_{corr}$\*>0.1 | % of stations corr ↓ and 0.05<$\Delta_{corr}$\*< 0.1 | % of stations corr ↓ and $\Delta_{corr}$\*>0.1 |
| ANN_SSM_NDVI | 3.8 | 0 | 2.86 | 0 | 9.95 | 5.97 |
| ANN_SSM_TEMP | 0 | 1.2 | 0 | 2.78 | 4.67 | 3.27 |
| ANN_SSM_EXP-FILT-T5 | 6.33 | 1.27 | 3.03 | 9.09 | 6.97 | 3.48 |
| ANN_SSM_EVAP-EFF-B60 | 10.39 | 1.3 | 0 | 2.94 | 6.74 | 5.7 |
| ANN_SSM_NDVI_EVAP-EFF-B60_EXP-FILT-T5 | 4.35 | 1.45 | 6.45 | 3.23 | 9.2 | 6.9 |

**Comment:** "Looking at Figure 4, I am impressed with the quality of the results. Using complex process models and measuring all the soil properties, I have never been able to simulate the water dynamics in different layers with such realism. I am impressed that a neural network trained with data from all over the world is able to reproduce with such fidelity the moisture levels between layers and the temporal variations in deep layers. No spurious variations are observed while the surface signal is particularly impacted by many rain events. If we can highlight the association of variables that allows such quality results, we have a major result for the understanding of water dynamics. This point must absolutely be highlighted."

**Reply:** As the reviewer mentioned, the ANN model was trained on a wide set of areas across the world having different climates and soil textures as described in (Souissi et al., 2020). This coverage across variable conditions contributes to making the model more generalizable. We didn't make a direct comparison of the outputs of the machine learning approach we used here to the predictions of complex physically-based models for water flow in variably saturated soils using for example the Richards-equations. We agree that it may be difficult to calibrate the pedotransfer functions for variably saturated flow in the soil and for the energy budget. This is mainly because of the sheer heterogeneity and unconsidered phenomena like hysteresis in the soil. This is also the main motivation of this paper.

In the revised paper, a paragraph has been added as follows: *«As shown in figure 4, the most complex ANN model is able to capture the variations of RZSM. This finding supports the added value of our hybrid approach based on an association of a machine learning method with process-related variables. Instead of injecting uncertain information, such as soil properties, in physical models, we used a nonparametric method that is related to physical processes without using forcing data going that may be subject to errors and potentially lead to inaccurate tracking of the long-term evolution of soil moisture.»*

Even though the results we obtain are of good quality over a given set of stations, this is not the case in all conditions. Time series of good and less good quality of fit have been added in appendix E for training, validation and test stations separately using models ANN_SSM and ANN_SSM_NDVI_EVAP-EFF-B60_EXP-FILT-T5 . Appendix E now reads:

*«Training stations:*

*Station 'Beloufoungou Mid' (network 'AMMA-CATCH')*

[Figure]

*Station 'HarmsWay' (network 'SCAN')*

[Figure]

*Station 'Cabrieres D'Avignon' (network 'SMOSMANIA')*

| *ANN_SSM* | *ANN_SSM_NDVI_EVAP-EFF-B60_EXP-FILT-T5* |
|---|---|
|
[Figure]
 | |

*Station 'Nephi' (network 'SCAN')*

| *ANN_SSM* | *ANN_SSM_NDVI_EVAP-EFF-B60_EXP-FILT-T5* |
|---|---|
|
[Figure]
 | |

*ISMN test stations*

*Station 'Wankama' (network 'AMMA-CATCH')*

| *ANN_SSM* | *ANN_SSM_NDVI_EVAP-EFF-B60_EXP-FILT-T5* |
|---|---|

[Figure]

Station '2.04' (network 'HOBE')

| ANN_SSM | ANN_SSM_NDVI_EXP-T5_EVAP-B60 |
|---|---|

[Figure]

We have enhanced the discussion of the results obtained in this study with respect to outputs from previous studies by adding the following paragraph:

*«Always in terms of the general performance of model ANN_SSM_NDVI_EVAP-EFF-B60_EXP-FILT-T5, about 75% of the stations have an RMSE less than 0.05 m³/m³ and around half of the stations have an RMSE less than 0.04 m³/m³. This accuracy is consistent, for instance, with the target value in SMAP (Entekhabi et al., 2010) and SMOS (Kerr et al., 2010) missions which is equal to 0.04 m³/m³ and also to the average sensor accuracy adopted by Dorigo et al. (2013) which is equal to 0.05 m³/m³. Overall, the most complex model ANN_SSM_NDVI_EVAP-EFF-B60_EXP-FILT-T5 can successfully characterize the soil moisture dynamics in the root zone since half of the stations have a correlation value greater than 0.7. Pan et al. (2017) developed different ANN models to estimate RZSM at depth of 20cm and 50cm over the continental United States using surface information. They found that half of the stations have RMSE less than 0.06 m³/m³ and more than 70% of stations have correlation above 0.7 when predicting RZSM at 20cm. However, the developed ANN was less effective in RZSM prediction at 50cm which is also in accordance with (Kornelsen and Coulibaly, 2014). In our study, the densest soil moisture network is 'SCAN', located in the USA. Soil moisture was predicted at a depth of 50cm over this network. Around half of the stations have a correlation value of above 0.6 and RMSE less than 0.04 m³/m³ after the integration of process-related inputs. Pan et al.,*

*(2017) suggests that the use of only time-dependent variables may not be sufficient for the ANN models to accurately predict RZSM and suggests adding soil texture data.»*

Also, the training experiments were designed to detect the impact of each process on the prediction quality. The association of variables is highlighted in figure (cf. figure 6).

Figure 4 shows the prediction results in Tunisia which sites fall into climate class 'Bsh'. Stations of similar climates, namely climate class 'Bsk' , were used in the training process. Table 5 was added in section 3.3 Robustness of the approach (please see Table 5 and updated section 3.3 above).

Also, time series over the Tunisian sites, with models ANN_SSM and ANN_SSM_NDVI_EVAP-EFF-B60_EXP-FILT-T5, were added in a bigger format in appendix G as follows:«

*Station Barrage-162 (Tunisia)*

| *ANN_SSM* | *ANN_SSM_NDVI_EVAP-EFF-B60_EXP-FILT-T5* |
|---|---|
|
[Figure]
 | |

*Station Barrouta_160 (Tunisia)*

| *ANN_SSM* | *ANN_SSM_NDVI_EVAP-EFF-B60_EXP-FILT-T5* |
|---|---|

[Figure]

*Station Bouhajla_164 (Tunisia)*

| ANN_SSM | ANN_SSM_NDVI_EVAP-EFF-B60_EXP-FILT-T5 |
|---|---|

[Figure]

*Station Hmidate_163 (Tunisia)*

| ANN_SSM | ANN_SSM_NDVI_EVAP-EFF-B60_EXP-FILT-T5 |
|---|---|

[Figure]

*Station P12 (Tunisia)*

[Figure]

*»*

**Comment:** "Finally, on the form, the article does not seem to me well written. Not being an English speaker, I find the English not very good and the text not always clear. An subtancial editing work is for me essential."

**Reply:** The submitted manuscript was reviewed by the AJE English editing service (the invoice is attached). The current submitted one has been greatly enhanced and modified and was thoroughly reviewed.

**Comment:** "In conclusion, I find the submitted draft article has not reached a stage of maturity allowing a publication. Some of the results are potentially extremely important and I therefore invite further analysis."

**Reply:** We thank the reviewer for taking the time to review our manuscript. We appreciate the suggestions for additional results and further analysis which helped us clarify some points and overall improve the quality of the paper. We believe that the aforementioned modifications will enrich the paper and we hope that all the reviewer's concerns have been addressed. More results and discussions were added to the revised paper. The appendices were also developed to include for instance more time series of good and less good quality of fit for training, validation and test stations separately using models ANN_SSM and ANN_SSM_NDVI_EVAP-EFF-B60_EXP-FILT-T5 (cf. appendix E).

**INVOICE**

**Invoice# SWPY3VY25**

**Balance Due**
**€0,00**

[Figure]

601 W Main St, Ste 102
Durham, North Carolina, 27701, USA
Tax ID: 412141424

| | |
|---|---|
| Invoice Date : | 01 Feb 2022 |
| Submission : | WPY3VY25 |
| Word Count : | 5613 |
| Title : | Integrating process-based information into ANN for root-zone soil moisture prediction |
| Discounts Applied : | $40 (BC690F) |

Bill To
**Mehrez Zribi**
CESBIO-CNRS, 18 Avenue Edouard Belin
Toulouse Cedex 9
31401
France
Mehrez Zribi

| # | Item & Description | Base Price | Discount | Amount |
|---|---|---|---|---|
| 1 | Advanced Editing
Advanced Editing | 386,76 | 35,48 | 351,28 |

| | |
|---|---|
| Sub Total | 351,28 |
| **Total** | **€351,28** |
| Payment Made | (-) 351,28 |
| **Balance Due** | **€0,00** |